



Atmospheric
Chemistry
and Physics

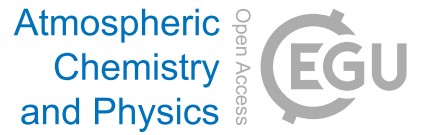

# Molecular characterization of organic aerosols in urban and forested areas of Paris using high-resolution mass spectrometry

Diana L. Pereira[1], Chiara Giorio[2], Aline Gratien[1], Alexander Zherebker[2], Gael Noyalet[1],
Servanne Chevaillier[3], Stéphanie Alage[3], Elie Almarj[1], Antonin Bergé[1,a], Thomas Bertin[3],
Mathieu Cazaunau[3], Patrice Coll[1], Ludovico Di Antonio[3], Sergio Harb[3,7], Johannes Heuser[3],
Cécile Gaimoz[3], Oscar Guillemant[3,4], Brigitte Language[3,b], Olivier Lauret[3], Camilo Macias[1],
Franck Maisonneuve[3], Bénédicte Picquet-Varrault[3], Raquel Torres[3,c], Sylvain Triquet[1], Pascal Zapf[3],
Lelia Hawkins[5], Drew Pronovost[5], Sydney Riley[5], Pierre-Marie Flaud[6], Emilie Perraudin[6],
Pauline Pouyes[6], Eric Villenave[6], Alexandre Albinet[7], Olivier Favez[7], Robin Aujay-Plouzeau[7],
Vincent Michoud[1], Christopher Cantrell[3], Manuela Cirtog[3], Claudia Di Biagio[1],
Jean-François Doussin[3], and Paola Formenti[1]

[1]Université Paris Cité and Univ Paris Est Creteil, CNRS, LISA, 75013 Paris, France
[2]Yusuf Hamied Department of Chemistry, University of Cambridge, Cambridge, CB2 1EW, United Kingdom
[3]Univ Paris Est Creteil and Université Paris Cité, CNRS, LISA, 94010 Créteil, France
[4]Université Paris-Saclay, Ecole Normale Supérieure Paris-Saclay,
4 avenue des sciences, 91190 Gif-sur-Yvette, France
[5]Department of Chemistry, Harvey Mudd College, 301 Platt Blvd, Claremont, California 91711, United States
[6]Univ. Bordeaux, CNRS, EPOC, EPHE, UMR 5805, 33600 Pessac, France
[7]INERIS, Parc technologique Alata BP2, 60550 Verneuil-en-Halatte, France
[a]now at: Laboratoire des Sciences du Climat et de l'Environnement, CEA-CNRS-UVSQ,
IPSL, Université Paris-Saclay, 91191 Gif-sur-Yvette, France
[b]now at: Unit for Environmental Sciences and Management,
North-West University, Potchefstroom, South Africa
[c]now at: Center for Research in Sustainable Chemistry-CIQSO, University of Huelva,
Campus de El Carmen, 21071, Huelva, Spain

**Correspondence:** Diana L. Pereira (diana.pereira@lisa.ipsl.fr) and Aline Gratien (aline.gratien@lisa.ipsl.fr)

**Abstract.** In order to study aerosols in environments influenced by anthropogenic and biogenic emissions to variable extents, $PM_1$ samples were collected during summer 2022 in the greater Paris area (ACROSS campaign, Atmospheric Chemistry Of the Suburban Forest, 14 June to 25 July) at two locations that represent the urban Paris and the suburban forested areas. They were analyzed using high-resolution mass spectrometry (HRMS) together with total carbon (TC) with a thermo-optical method. Both sites are compared here to explore differences in aerosol composition from urban and forested environments. The TC analysis shows similar organic carbon (OC) concentrations at both sites ($3.2 \pm 1.8\,\mu g\,m^{-3}$ for Paris and $2.9 \pm 1.5\,\mu g\,m^{-3}$ for Rambouillet) and higher elemental carbon (EC) values in the urban area. Both OC and EC concentrations did not show significant variations for daytime and nighttime conditions. This work highlights the influence of anthropogenic inputs on the chemical composition of urban and forested areas, derived from the presence of CHO and CHON compounds but also the detection of two sulfur-containing compounds ($C_5H_{12}SO_7$ and $C_{10}H_{17}NSO_7$), which could be tentatively assigned as organosulfates. A smaller number of aromatic compounds were observed for clean periods that better represent the local biogenic and anthropogenic contributions in Rambouillet and Paris, respectively.

Published by Copernicus Publications on behalf of the European Geosciences Union.

## 1 Introduction

Organic aerosols (OAs) represent an important fraction of the fine aerosol mass (up to 90 %) (Chen et al., 2022; Kanakidou et al., 2005) that can impact the Earth's climate through their interactions with clouds (Andreae and Rosenfeld, 2008; IPCC, 2021; Rosenfeld et al., 2014), solar radiation (Haywood, 2016), and air quality (Chen and Kan, 2008). Despite the importance of these particles, their composition and formation processes are not fully understood, and gaps remain in their chemical characterization and description (Akinyoola et al., 2024; Kalberer, 2015). Different environments provide different aerosol sources with varied chemical composition and influence on the atmosphere. Megacities such as Paris, Mexico City, Beijing, and New York are known for their high populations and local anthropogenic emissions that contribute to the atmospheric particulate matter (PM) levels (Karagulian et al., 2015; Cheng et al., 2016), while remote environments such as forested areas and oceans contribute mostly biogenic emissions (Shen et al., 2015; Zhu et al., 2016). At their interfaces, urban and remote environments can be affected by mixtures of both biogenic and anthropogenic emissions, which influence OA formation and composition (Rattanavaraha et al., 2016; McFiggans et al., 2019; Shrivastava et al., 2019). The enhancement of biogenic aerosol formation under the influence of anthropogenic pollutants has already been reported (Bryant et al., 2023; Rattanavaraha et al., 2016; Shrivastava et al., 2019; Yee et al., 2020); however, the opposite effect for specific mixtures such as isoprene, CO, or $CH_4$ with $\alpha$-pinene was also observed (McFiggans et al., 2019), highlighting the complexity of OA formation in mixed environments.

Complex and simultaneous physicochemical processes influence aerosol formation and growth (Hallquist et al., 2009) in the atmosphere. Therefore, intensive and long-term field observations using combinations of online and offline techniques have been performed to gain insights into OA chemical composition, source apportionment, properties, and possible implications on the atmospheric processes (Molina et al., 2010; Bressi et al., 2013; Zhang et al., 2013; Artaxo et al., 2017; Cantrell and Michoud, 2022). These techniques include online aerosol mass spectrometers (Zhang et al., 2007), total carbon content with semi-continuous carbon analyzers (Karanasiou et al., 2020) or offline thermo-optical techniques (Cao et al., 2005; Ma et al., 2016), and chemical composition of samples collected on filters (Yan et al., 2009; Ding et al., 2012; Michoud et al., 2021) particularly using chromatographic techniques coupled to mass spectrometers (Kourtchev et al., 2013). High-resolution mass spectrometers (e.g., Orbitrap) have been shown to provide interesting insights into OA in urban, suburban, and/or remote areas (Kourtchev et al., 2014; Daellenbach et al., 2019; Giorio et al., 2019; Wang et al., 2022; Amarandei et al., 2023). For

example, a strong biogenic influence at a urban background and two remote locations in the Alpine valleys in Switzerland was observed during summertime (Daellenbach et al., 2019). A high contribution of saturated oxidized compounds together with the presence of organosulfates suggested that secondary organic aerosol (SOA) formation from biogenic volatile organic compound (VOC) precursors plays an important role during the summer even in urban areas (Giorio et al., 2019; Amarandei et al., 2023). The importance of anthropogenic oxidized compounds has also been highlighted, with a strong contribution from traffic emissions in urban areas (Kourtchev et al., 2014; Wang et al., 2022).

The Paris area (comprising Paris city and its suburbs) is a relatively compact urban zone of ca. 40 km by 40 km surrounded by low, urbanized areas mostly composed of intensive agriculture fields and forest. Mixing between anthropogenic and biogenic emissions can occur, especially when the plume of Paris travels away from the city. This encounter can be favored by anticyclonic weather that allows the interaction between air masses due to slow-moving conditions (Lagmiri and Dahech, 2023; Wei et al., 2011). Anticyclone conditions have previously led to the identification of PM accumulation (Beekmann et al., 2015) and variability of aerosol particles (Bressi et al., 2013) in the Paris region. Previous studies in this area involving offline analysis of the aerosol chemical composition have been focused on total carbon (TC) determination (Favez et al., 2009), combined with ion chromatography analysis (Hodzic et al., 2006; Gros et al., 2007; Sciare et al., 2010; Bressi et al., 2013), and only a few reported analyses at the molecular scale (Lanzafame et al., 2021; Srivastava et al., 2018, 2019). Gros et al. (2007) reported traffic pollution as an important aerosol source in the Paris urban area as seen from the measurement site located at the Paris City Hall during the spring period. Analyses performed the summer at the suburban area of Saclay (25 km southwest from Paris) showed the predominance of primary aerosols and highlighted the contribution of both anthropogenic and biogenic sources (Hodzic et al., 2006). Previous measurements in central Paris showed the influence of residential wood-burning emissions in the winter (Favez et al., 2009) and highlighted the local contributions for the primary fraction of the aerosol and continental photochemical-aged air masses during the spring (Sciare et al., 2010).

All of these studies were mostly based on a single sampling point, with the exception of Bressi et al. (2013), who compared the chemical composition of a set of samples collected at five sites of the AIRPARIF air quality stations during 1-year measurements (September 2009 to September 2010). Those sites included one urban location in the Paris center (fourth district); one suburban station (10 km northeast of Paris center station); and three rural sites located 65 km northeast, 50 km northwest, and 60 km south of the urban station. Based on ion analysis and TC measurements only,

a spatial homogeneity of the aerosol chemical composition was observed during that period for the sites, with higher levels of local anthropogenic contributions as the sites get closer to the Paris center. Those studies provided a base for the typology of aerosol sources with a limited ion spectrum ($NO_3^-$, $SO_4^{2-}$, $Na^+$, $NH_4^+$, $K^+$, $Mg^{2+}$, and $Ca^{2+}$) analyzed. This, together with similarities between sites observed for long measurement periods and the lack of seasonal tendencies, raises the concern that the description of the inorganic aerosol fraction and organic carbon (OC) measurements cannot properly capture the full chemical process occurring. Recent studies focus on measurements of the OA at the suburban SIRTA (Site Instrumental de Recherche par Télédétection Atmosphérique; Haeffelin et al., 2005) site (Lanzafame et al., 2021; Srivastava et al., 2018, 2019), located to the southwest of the Paris center (25 km). Influence of secondary processes in the aerosol composition from the early spring was observed by Srivastava et al. (2018, 2019). The temporal variability of pinene, isoprene, $\beta$-caryophyllene, anthropogenic SOA acids, and nitroaromatic markers was investigated, showing a seasonal dependence of the processes enlightened by nitroaromatic and isoprene markers (Lanzafame et al., 2021).

The aforementioned studies showed the contribution of different local urban sources mainly associated with traffic emissions, especially during the summer, together with contributions of mid- and long-range-transport biogenic and anthropogenic species, of primary or secondary origin. They additionally highlighted the effect of the seasonal variability for different specific species associated with emission factors and seasonal meteorological conditions. The studies performed to understand the atmospheric chemical composition of rural, suburban, and urban areas in Paris provide important information on the major chemical fractions, carbon content, and some molecular tracers. However, information on the OA chemical composition and its temporal and spatial variability is still missing. Building upon the existing knowledge, our work aims at providing new descriptions of the molecular composition and day–night variability of the OA fraction from simultaneous measurements in two urban and forested environments in the Paris region in summer 2022 from measurements performed in the framework of the ACROSS (Atmospheric Chemistry Of the Suburban Forest) intensive campaign.

## 2  Methods

Atmospheric measurements were performed during the ACROSS campaign, which aims to understand the mixing between the biogenic and anthropogenic emissions and their impact on aerosol formation and aging. In the Paris urban area, anthropogenic compounds can be emitted and exported to the forested areas, interacting with their local emissions. Therefore, during ACROSS, measurements were performed at ground-based, airborne, and space-based platforms located in different urban, semi-urban, and rural locations in the greater Paris area. Further details on the sites and campaign description are provided in Cantrell and Michoud (2022). In this work, we focus on the aerosol chemical composition measurements performed at ground level at two locations that represent the urban Paris and the suburban forested areas.

### 2.1  Ground level sampling

Atmospheric sampling was performed in summer 2022 (14 June–25 July) on the seventh-floor terrace of the Lamarck B building at the Université Paris Cité, located 20 m above ground level (m a.g.l.) (48.8277° N and 2.3806° E; henceforth referred to as Paris) and at ground level in the Rambouillet forest (48.6866° N and 1.7045° E; henceforth referred to as Rambouillet).

Paris city, with a population of $\sim$ 2 million inhabitants (Bilan démographique, 2024) and about 12 million in the entire greater Paris area, is characterized by urban local emissions from traffic and aerosol contributions from mid- and long-range transport (Beekmann et al., 2015; Bressi et al., 2013, 2014). On the contrary as observed in Fig. 1, the Rambouillet site is a dense forest area located 43 km from the center southwest bound of Paris, far from local anthropogenic contributions and susceptible to the influence to urban plume arrival from the northeast. The Rambouillet forest mainly consists of oaks and pine trees (Office National des Forêts, 2023) and has an extension of 14 000 ha.

### 2.1.1  Filter sampling

Aerosol filter sampling of the $PM_1$ fraction was performed during day (06:00–22:00 local time, LT) and night (22:00–06:00 LT) on 150 mm diameter quartz fiber filters (Pallflex Tissuquartz). Samples were collected using the automatic continuous high-volume aerosol sampler (30 $m^3 h^{-1}$) DHA-80 (DIGITEL Enviro-Sense) equipped with a $PM_1$ sampling head, directly exposed to the ambient air. Sampling times were selected to account for different daily processes in the presence of sunlight (daytime) and in the absence of light (nighttime). Therefore, sampling performed during the day included an important fraction of anthropogenic activities mainly associated with traffic contribution during rush hours between 08:00–10:00 and 20:00–22:00 LT, as previously reported in the Paris urban area (Gros et al., 2007; Sciare et al., 2010). Prior to sampling, the quartz fiber filters were conditioned at 550 °C for 8 h and conserved in pre-baked aluminum foil, sealed in plastic bags. After exposure, samples were conserved in pre-baked and sealed aluminum foils at −20 °C, then transported to the laboratory, where they were punched to smaller fractions (∅ = 30 and 46 mm) for chemical analysis. The smaller fractions were also conserved following the same protocol.

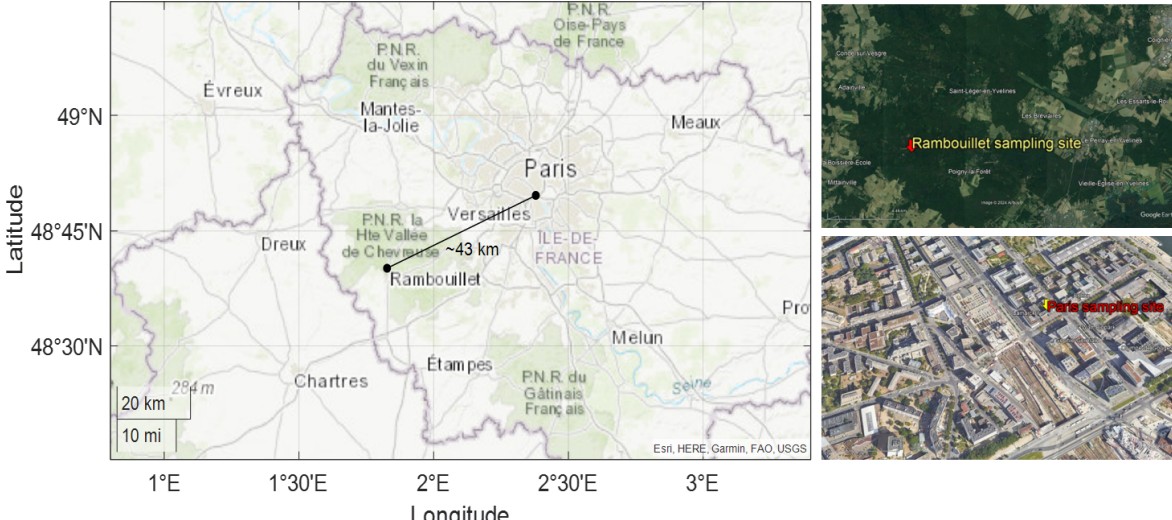

**Figure 1.** Sampling sites in the Paris region for aerosol sampling in the ACROSS campaign during summer 2022. The map on the left shows the spatial disposition between sites (The MathWorks Inc., 2022). The urban area sampling site is located at the Université Paris Cité, and the forested one is in the Rambouillet forest, displayed on the right side (© Google Earth).

### 2.1.2   Additional data

Meteorological parameters such as relative humidity (RH), temperature, and wind speed and direction for the Rambouillet site were provided by the Centre National de Recherches Météorologiques (CNRM) through the Meteo France mobile facility (Denjean, 2023). Additional data of $NO_x$, $O_3$, and $SO_2$ concentrations for Rambouillet were collected by monitors from the PortablE Gas and Aerosol Sampling UnitS (PE-GASUS) mobile facility (Giorio et al., 2022). At PEGASUS, gas measurements were performed using Teflon tubes with 6.35 mm diameter with the line inlet placed at the top of the container at 2 m a.g.l. A $NO_x$ APNA370 monitor (Horiba) based on $O_3$ chemiluminescence, an $O_3$ APOA370 monitor (Horiba) based on UV absorption, and a $SO_2$ APSA370 monitor (Horiba) based on UV fluorescence were utilized. At the Paris site, we used a Lufft WS600 meteorological station (Di Antonio et al., 2023), a $NO_x$ AC32M monitor (Environment SA) based on $O_3$ chemiluminescence, an $O_3$ 41M monitor (Environment SA) based on UV absorption, and a $SO_2$ AF22 monitor (Environment SA) based on UV fluorescence. For Paris, gas sampling was performed at the top of the building (30 m a.g.l) through a 12 m long Teflon tube, with a 17.5 mm inner diameter at $40\,L\,min^{-1}$, to a glass manifold where all gas phase instruments sampled ambient air.

### 2.2   Chemical analysis

Organic carbon (OC) and elemental carbon (EC) were measured using a thermo-optical analyzer (Sunset Laboratory Inc.) on a $1.5\,cm^2$ filter surface following the EUSAAR2 protocol (Cavalli et al., 2010). The Sunset analyzer was calibrated using a sucrose (purity $> 99.5\,\%$) solution on a

$1.5\,cm^2$ filter surface at concentrations between 0.42 and $40\,\mu g\,C\,cm^{-2}$, with a limit of detection of $0.25\,\mu g\,C\,cm^{-2}$ and a limit of quantification of $0.42\,\mu g\,C\,cm^{-2}$. Prior to each analysis, an instrumental blank and a point at $10\,\mu g\,C\,cm^{-2}$ were measured as a quality control. OC and EC values were automatically calculated with the software OCBC835 (Sunset Laboratory), and the split point was manually verified to ensure proper assignation. Uncertainties in the measurements were obtained by considering the 5 % of the carbon concentration plus $0.1\,\mu g\,C\,cm^{-2}$ as the minimum instrumental error, as suggested by the manufacturer (Sunset Laboratory).

The chemical compositions of OA extracts were studied using high-resolution mass spectrometry (HRMS). Filters of 46 mm diameter were placed in pre-cleaned glass vials and extracted three times in 3 mL methanol (LC-MS grade, Fisher Scientific) by 30 min sonication in slurry ice. Methanol was used due to its suitability for the extraction of polar unsaturated compounds (Zherebker et al., 2024) and high extraction efficiency of OA (Giorio et al., 2019), enabling analysis without an extra purification step. Extracts were combined for each sample and filtered with 0.45 and $0.2\,\mu m$ syringe filters (Iso-Disc® PTFE filters, Ø 4 mm) consecutively. The solvent was partially evaporated using a gentle stream of nitrogen to $400\,\mu L$. The final solutions were stored at $-18\,°C$ prior to analysis. Additional details on the extraction protocol can be found in Kourtchev et al. (2014).

Non-target mass spectrometric analysis was performed using an LTQ Orbitrap mass spectrometer (Thermo Scientific) equipped with a chip-based nanoESI source (TriVersa Nano-Mate, Advion) operating in negative ionization mode. The source parameters were a gas pressure of 0.30 psi, a spray voltage of $-1.4\,kV$, and an injection volume of $7\,\mu L$. Na-

noESI source was used to achieve higher ionization efficiency for a variety of analytes (Kourtchev et al., 2014). All spectra were recorded in triplicate with a 1 min data acquisition time, with a nominal resolution of 100 000 and in two scan ranges (50–500 $m/z$ and 150–1000 $m/z$). Due to the solvent selection and the use of the ESI source, analyses at molecular scale focus on the polar fraction of the methanol-soluble OA.

## 2.3 HRMS data processing

HRMS data treatment was conducted following Zielinski et al. (2018). Peaks were extracted and assigned using Xcalibur 2.1 (Thermo Scientific) with a mass tolerance of 4 ppm. Atomic constraints for formulae assignment were $^{14}N$ (0–5), $^{16}O$ (0–50), $^{12}C$ (1–100), $^{1}H$ (1–200), $^{32}S$ (0–2), $^{34}S$ (0–1), and $^{13}C$ (0–1). Formula lists and peak intensities were further processed including internal calibration, noise removal, blank subtraction, and additional atomic constraints. The latest include elemental ratios set as $0.3 \leq H/C \leq 2.5$, $O/C \leq 2$, $N/C \leq 1.3$, $S/C \leq 0.2$, $^{13}C/^{12}C \leq 0.011$, and $^{34}S/^{32}S \leq 0.045$, double-bond-equivalent (DBE) values, nitrogen rule, and isotopic filtering (Zielinski et al., 2018). Formulae with the smallest absolute mass errors were used for multiple assignments after mass shift correction, and only formulae found in all triplicates were considered. The formulae assigned were grouped based on the molecular composition as CHO, CHON, CHOS, CHONS, CHNS, CHN, and CHS families. However, due to the higher ionization efficiency in the negative mode, this work focuses on O-containing compounds (CHO, CHON, CHONS, and CHOS). This selection is also supported by the predominance of oxygenated compounds on the OA on the $PM_1$ fraction reported in different urban and rural areas (Zhang et al., 2007).

After formula assignment, van Krevelen (VK) diagrams (Kim et al., 2003) were used to visualize the differences between the samples from Paris and Rambouillet. Following Kourtchev et al. (2013), three compound domains such as aliphatic, low-oxidation aromatic, and more oxidized aerosol were used herein. The aliphatic group was attributed at low O/C ratios ($< 0.5$ TS1) and high H/C ($> 1.5$ TS2), low-oxidation aromatic domain was present at O/C $< 0.5$ TS3 and H/C $> 0.5$ TS4, and more oxidized compounds could be found at higher O/C ($> 0.5$). More details are presented in the Supplement.

The aromaticity equivalents (Xc) of the aerosol samples of Rambouillet and Paris were calculated following Yassine et al. (2014) with Eq. (1):

$$Xc = \frac{3[DBE - (mO + nS)] - 2}{DBE - (mO + nS)}. \tag{1}$$

The $m$ and $n$ parameters designate the fractions of O and S atoms involved in the $\pi$ bonds. In order to provide a conservative estimate of unsaturation that is accounted for in only carbon–carbon $\pi$ bonds, we used $m$ and $n = 0.5$, which correspond to the maximum possible fraction of $sp^2$-hybridized oxygen atoms as carboxyl groups. While the conservative approach provides only a lower boundary for the contribution of highly unsaturated compounds (tentatively assigned as aromatic and condensed compounds), a predominance of carboxylic acid species over other functionalities is expected for analysis in negative ion mode (Kourtchev et al., 2016). The role of carboxylic acids as one of the major fractions of aerosol particles was observed during the summer for field samples in European cities in Corsica (Michoud et al., 2021) and the Czech Republic (Horník et al., 2020). Although this assumption can lead to an underestimation of possible aromatic fraction, we considered $m$ and $n = 0.5$ as previously applied in the literature (Tong et al., 2016; Yassine et al., 2014). DBE represents the degree of unsaturation of a compound (Wozniak et al., 2008), and it is computed considering the number of C, H, and N atoms as follows:

$$DBE = 1 + \frac{1}{2}(2C - H + N). \tag{2}$$

Yassine et al. (2014) proposed threshold values of Xc, where $Xc \geq 2.5$ accounts for aromatics and $Xc \geq 2.71$ for condensed aromatics. Following this classification, the Xc values are calculated and grouped here as unsaturated ($Xc < 2.5$), aromatic ($2.5 \leq Xc < 2.71$), and condensed aromatic ($Xc \geq 2.71$) compounds. Aromatics may contain one benzene ring, while condensed aromatics can contain fused rings.

## 2.4 Statistical analysis

Pearson correlation and pairwise cosine distances were calculated to analyze the existing relationships between chemical composition (OC and EC concentrations, number of formulae), meteorological conditions, and anthropogenic contaminants observed during the campaign. Atmospheric conditions used for correlation analysis are reported in Table S1 in the Supplement, and they represent average data observed during the sampling period.

## 3 Results and discussion

### 3.1 Variability of OC and EC concentrations

The OC and EC concentrations observed during summer 2022 in Paris and Rambouillet are summarized in Fig. 2. In Paris, OC concentrations ranging from 0.7 to 10.0 µg m$^{-3}$ and maximum EC concentrations of 1.3 µg m$^{-3}$ were observed between 14 June and 25 July. A shorter dataset is available for Rambouillet, between 27 June and 22 July, with OC concentrations varying between 0.8 and 7.7 µg m$^{-3}$. The OC time series of Paris showed higher concentrations at the beginning of the campaign (14 and 24 June) and after 12 July, while for Rambouillet there was an increasing trend over

time which coincides temporally with the increased OC concentration observed from the end of June to July for Paris. Wind direction showed a north to east (0 to 75°) predominant contribution at the beginning of the period (before 24 June) and additional contributions from western (230 to 355°) air masses for Paris. In Rambouillet, a similar behavior was observed at the beginning of July, while a transition from northeast to west was observed for the rest of the period. Wind directions from the northeast (between 0 and 90°) suggests the possible impact of Paris emissions into the forest during the second week of July from 7 to 13 and 15 to 16 July. Higher wind speed values were observed over the forest site (up to 7.1 m s$^{-2}$) than for downtown Paris (up to 4.3 m s$^{-2}$).

The temporal variability of RH, temperature, and concentrations of NO$_x$, O$_3$, and SO$_2$ at both sites is also shown in Fig. 2. Mean values of RH of 51.7 % (min 16.0 % and max 93.0 %) and temperature of 22.5 °C (min 13.3 °C and max 39.5 °C) were observed for the Paris site. In Rambouillet, mean values of 60.6 % (min 14.4 % and max 96.7 %) and 19.8 °C (min 6.0 °C and max 38.9 °C) were reported for RH and temperature, respectively. Higher values of NO$_x$ were observed in Paris (mean values of 10.7 and 2.2 ppb with maximum concentrations of 90.9 and 15.2 ppb for Paris and Rambouillet, respectively), while O$_3$ concentrations followed a similar profile at both sites with higher concentrations in Rambouillet of 6.0 ppb on average. Low SO$_2$ concentrations were observed for both sites with values being generally below 1 ppb, but some peaks were also detected with maximum values of 8.5 and 6.5 ppb for Paris and Rambouillet, respectively. Averages over the filter sampling times for the parameters described are presented in Table S1.

The difference between the forested and urban environments is observed looking into the EC concentrations, which are higher in Paris due to the densely populated area, which translates into higher local anthropogenic activities. In contrast, similarities between the profiles of the OC concentrations for the common period between both sites (27 June to 22 July) may suggest common organic sources influencing the chemical composition. Given that different wind directions are observed after 8 July, these similarities can be influenced by local sources and transported air masses. Previous summer measurements (Gros et al., 2007) of OC in the Paris urban area highlight local traffic contributions. As observed at both sites, the increase in the OC concentrations at the end of July was accompanied by an increase in temperature, NO$_x$, and O$_3$, which can enhance the chemistry and derive into a higher OA formation (Luo et al., 2021; Shrivastava et al., 2019; Xu et al., 2022). Further influence of the meteorological parameters and anthropogenic pollutants on the OC concentrations and composition are explored in Sect. 3.3.

In this study, mean OC concentrations of $3.2 \pm 1.7$ and $2.9 \pm 1.5$ µg m$^{-3}$ were observed for the whole sampling period in Paris and Rambouillet, respectively. Such values are around the annual local/regional averages previously found in PM$_{2.5}$ by Bressi et al. (2013), who reported 3.0 µg m$^{-3}$

in the urban center of Paris, 3.2 µg m$^{-3}$ at a suburban site (10 km from Paris center), and between 2.1 and 2.9 µg m$^{-3}$ in the rural areas in the north and south of Paris, located > 50 km from the center. Concentrations in the present analysis are also in line with the average summertime OC concentration of 2.9 µg m$^{-3}$ at the suburban (southwest of Paris center) SIRTA site reported by Lanzafame et al. (2021).

Mean OC and EC concentrations for Rambouillet observed in this study fall in a similar range to mean values of 3.6 and 0.1 µg m$^{-3}$ reported in the rural area of Hyytiälä (Finland) for the PM$_1$ fraction (Daellenbach et al., 2019) and 4.2 and 0.2 µg m$^{-3}$ for the PM$_{2.5}$ in the forest on the Great Hungarian Plain (Hungary) (Ion et al., 2005; Kourtchev et al., 2009). Our average OC and EC concentrations in Paris are of the same order of magnitude as those measured in the urban area of Seoul (South Korea), where OC and EC concentrations of 3.5 and 1.6 µg m$^{-3}$ for PM$_{2.5}$ were reported (Yoo et al., 2022). While our mean OC concentration in Paris is of the same magnitude, the EC concentration in Seoul is 4 times higher (Yoo et al., 2022) than the observed in this study (0.4 µg m$^{-3}$).

## Diurnal variations of OC and EC concentrations

For the periods of data overlap for the two sampling sites (27 June to 22 July), Fig. 2 shows a similar trend for the OC concentrations for most of the days with close mean concentrations values for OC of $3.4 \pm 1.8$ µg m$^{-3}$ in Paris and OC of $2.9 \pm 1.5$ µg m$^{-3}$ in Rambouillet (Table 1) and good OC concentration correlations between both sites ($r > 0.70$, $p$ value $< 0.05$) as shown in Fig. S1. Additionally, at correlating OC concentrations during daytime in Paris and nighttime in Rambouillet, the moderate positive correlation value of 0.55 ($p$ value $= 0.02$) suggests that some organic compounds could be formed in the urban area and transported to the forested one, influencing the chemical composition of the consecutive filter.

Measured OC and EC concentrations were compared for both sites in Table 1 for day and night periods. Higher EC concentrations were observed for Paris than for Rambouillet with maximum daytime values of 1.3 µg m$^{-3}$ and nighttime values of 1.2 µg m$^{-3}$. Higher EC concentrations for Paris highlight the urban nature of the site, while EC detection was not expected in Rambouillet, given its forested nature. In Paris, mean OC concentrations of 3.1 and 3.7 µg m$^{-3}$ were observed during day and night, while in Rambouillet mean values of 2.8 and 3.1 µg m$^{-3}$ were observed. Non-significant variability was observed between day and night for OC and EC concentrations. The proximity of the values for day and night concentrations observed at both sites (Table 1) and the similar trends (Fig. 2), together with the good correlations observed for OC concentrations (Fig. S1), may suggest common sources and/or air masses affecting the OA composition of both sites.

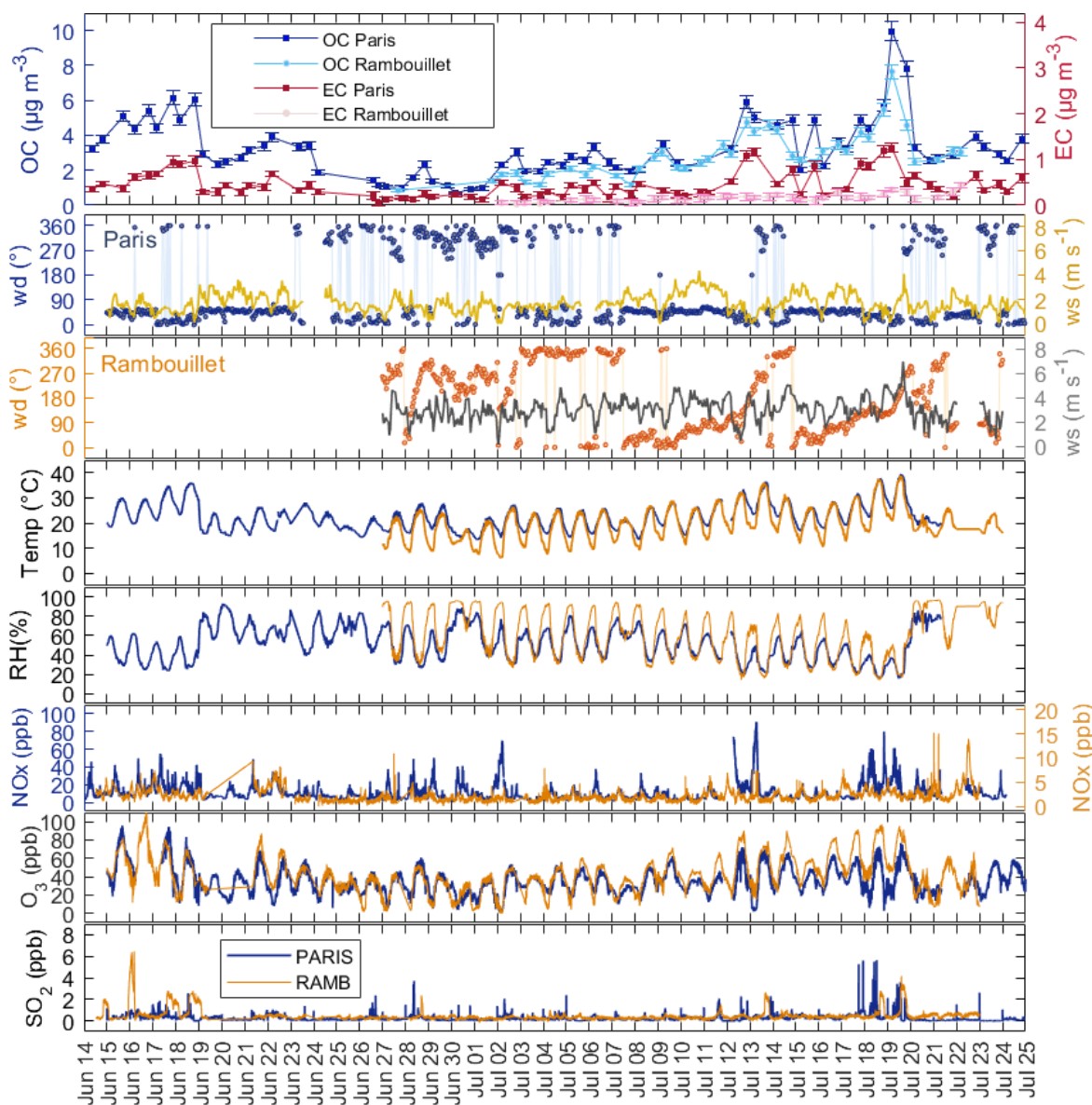

**Figure 2.** Temporal series of the OC, EC, NO$_x$, O$_3$, and SO$_2$ concentrations and temperature, RH, and wind conditions observed during summer 2022 between 14 June and 25 July. Data for OC (blue) and EC (red) for Paris and Rambouillet are reported with their standard deviations represented by the error bars, accounting for sample density variations and instrumental errors. Wind conditions are presented by wind speed (yellow and gray) and wind direction (blue and orange) and reported here for hourly average of daily measurements. Measurements for NO$_x$, O$_3$, and SO$_2$ for Paris (blue) and Rambouillet (orange) were performed at 30 m a.g.l. in Paris and at ground level in Rambouillet.

**Table 1.** Summary of OC and EC concentrations observed in Paris and Rambouillet during summer 2022. Maximum, minimum, and mean concentrations are reported for the total data collected from 14 June to 25 July. Day and night values of the concentrations are reported for the period of data overlap for the two sites (27 June to 22 July). The mean concentrations are reported with their standard deviation along the time series.

| | Paris | | Rambouillet | |
|---|---|---|---|---|
| | OC (µg m$^{-3}$) | EC (µg m$^{-3}$) | OC (µg m$^{-3}$) | EC (µg m$^{-3}$) |
| All data – mean (min–max) | $3.2 \pm 1.7$ (0.7–10.0) | $0.4 \pm 0.3$ (0.1–1.3) | $2.9 \pm 1.5$ (0.8–7.7) | $0.2 \pm 0.1$ (0.0–0.4) |
| Overlap period – mean (min–max) | $3.4 \pm 1.8$ (1.0–10.0) | $0.5 \pm 0.3$ (0.1–1.3) | $2.9 \pm 1.5$ (0.8–7.7) | $0.2 \pm 0.1$ (0.0–0.4) |
| Daytime – mean (min–max) | $3.1 \pm 1.8$ (1.0–10.0) | $0.5 \pm 0.3$ (0.1–1.3) | $2.8 \pm 1.5$ (0.8–7.7) | $0.2 \pm 0.1$ (0.1–0.4) |
| Nighttime – mean (min–max) | $3.7 \pm 1.7$ (0.9–7.8) | $0.5 \pm 0.4$ (0.2–1.2) | $3.1 \pm 1.6$ (1.2–5.5) | $0.2 \pm 0.1$ (0.0–0.3) |

Temperature variations between day and night periods associated with the solar irradiance together with higher ozone concentrations observed at day for both sites (Fig. 2) highlight the influence of different oxidation processes influencing the OC concentrations during the day. Higher temperature observed during the day can also enhance SOA precursor emissions such as monoterpenes (Bourtsoukidis et al., 2024; Malik et al., 2023), which then influence photochemical OA formation (Lin et al., 2009). At night, lower temperatures may favor the transition to the particle phase (Cahill et al., 2006; Giorio et al., 2019), and nitrate radical chemistry can affect SOA formation (He et al., 2021), increasing the OC concentrations. In addition to the influence of temperature, the planetary boundary layer (PBL) dynamics can influence OC and EC concentrations. During the day, the boundary layer height is deeper due to solar heating, which influences the mixing and dilution of pollutants. On the other hand, during the night the PBL becomes shallow, causing a nocturnal stability, which can enhance pollutant concentrations (Li et al., 2021; Wang et al., 2023; Zhang et al., 2020). Therefore, the lack of tendencies for OC and EC concentrations between day and night can result from a combination of different atmospheric processes.

Specific samples were selected to further understand differences in the aerosol chemical composition using the OC concentration and pollutant concentrations as indicators of chemistry. Samples from 3 and 4 July showed OC concentrations below the average (i.e., $3.2\,\mu g\,m^{-3}$ for Paris and $2.9\,\mu g\,m^{-3}$ for Rambouillet), further considered the background, while samples from 11 to 13 and 17 to 19 July (Fig. 2) were representative of polluted conditions as their concentrations were higher than the mean value. Such OC variations were consistent with higher concentrations of $O_3$, $NO_x$, and $SO_2$ for the pollution periods compared to the background one (Fig. 2). Backward trajectories were calculated for 24 h for the Rambouillet site with the HYSPLIT model (Stein et al., 2015) using the Global Data Assimilation System (GDAS) at 1° resolution. This highlights the contribution of different air masses during the background and pollution periods. As observed in Fig. S2, during the background period (3 July), air masses arriving from the northwest show maritime and continental contributions. During the pollution period (12 July), air masses arrive from the northeast, crossing the densely populated urban areas of Belgium and Paris city. During the second period of pollution (17 to 19 July), higher temperatures, OC concentrations, and an atypical event of long-range transport of biomass-burning emissions from the south of France on 19 July were also reported (Menut et al., 2023). In agreement, back-trajectory analysis highlighted the arrival of air masses from the south of Paris, showing that inside the second period, the aerosols may be influenced by different chemical processes and sources.

## 3.2 Molecular composition

The HRMS analyses for day and night samples from the urban and forested area of Paris were compared to investigate differences and similarities in the particle chemical composition at both sites. The mass spectra of specific samples representing the background (3 and 4 July) and polluted periods (11 to 13 and 17 to 19 July) are summarized in Fig. 3. Between 1639 and 5040 elemental formulae were identified with molecular weights mostly distributed below 600 $m/z$. It is important to highlight the lower number of molecular formulae assigned in the background samples of 3 July for the Rambouillet site, with 1639 and 2076 formulae for day and night samples (Table S1).

Considering the analyses performed on different urban and rural environments, we suggest possible compounds that can be associated with the molecular formulae identified in this study; however, they could also be associated with other isomers. A predominant signal observed at 215.023 $m/z$ in all samples for $C_5H_{12}SO_7$ is compatible with an isoprene oxidation product as suggested by Zherebker et al. (2024). $C_{10}H_{17}NSO_7$ formula at 294.065 $m/z$ is compatible with an $\alpha$-pinene oxidation product formed under the presence of $NO_x$ and $SO_2$ (Surratt et al., 2007, 2008), showing a higher relative intensity during the night period in Paris. This organosulfate has been previously detected in the field (Kourtchev et al., 2014, 2016; Giorio et al., 2019; Wang et al., 2022) and may highlight the influence of anthropogenic pollutants on SOA formation when mixed with biogenic compounds as it is form in the presence of the anthropogenic pollutants $NO_x$ and $SO_2$. The dependence on organosulfate formation in the presence of anthropogenic species has been followed during the oxidation of monoterpenes and isoprene. The presence of acidic sulfate seeds (Iinuma et al., 2009; Surratt et al., 2007) and additional presence of $NO_x$ species for the nitrooxy types (Surratt et al., 2008) were observed. The mechanisms proposed for organosulfate formation involve sulfate species reactions via epoxide formation, esterification, aqueous reactions, or direct reaction with gaseous $SO_2$ (Brüggemann et al., 2020). On the other hand, nitrate radical chemistry was suggested to determine nitrooxy organosulfate formation (Iinuma et al., 2007). It is important to consider that, in Rambouillet, $NO_x$ and $SO_2$ levels and sulfur compounds affecting this chemistry can also be influenced by microbial activity (Andersen et al., 2024). Although we cannot attribute the molecular formula $C_7H_{12}SO_{11}$ at 303.003 $m/z$, the fact that it only appears in Rambouillet samples may suggest a biogenic origin.

Peaks 133.014 $m/z$ ($C_4H_6O_5$), 147.030 $m/z$ ($C_5H_8O_5$), 167.071 $m/z$ ($C_9H_{12}O_3$), 171.066 $m/z$ ($C_8H_{12}O_4$), 187.061 $m/z$ ($C_8H_{12}O_5$), and 203.057 $m/z$ ($C_8H_{12}O_6$) appeared at high relative intensities in all samples. These CHO compounds may have anthropogenic or biogenic origins (Kourtchev et al., 2014). For example, $C_8H_{12}O_4$ and $C_8H_{12}O_6$ are compatible with terpenylic acid and 3-methyl-

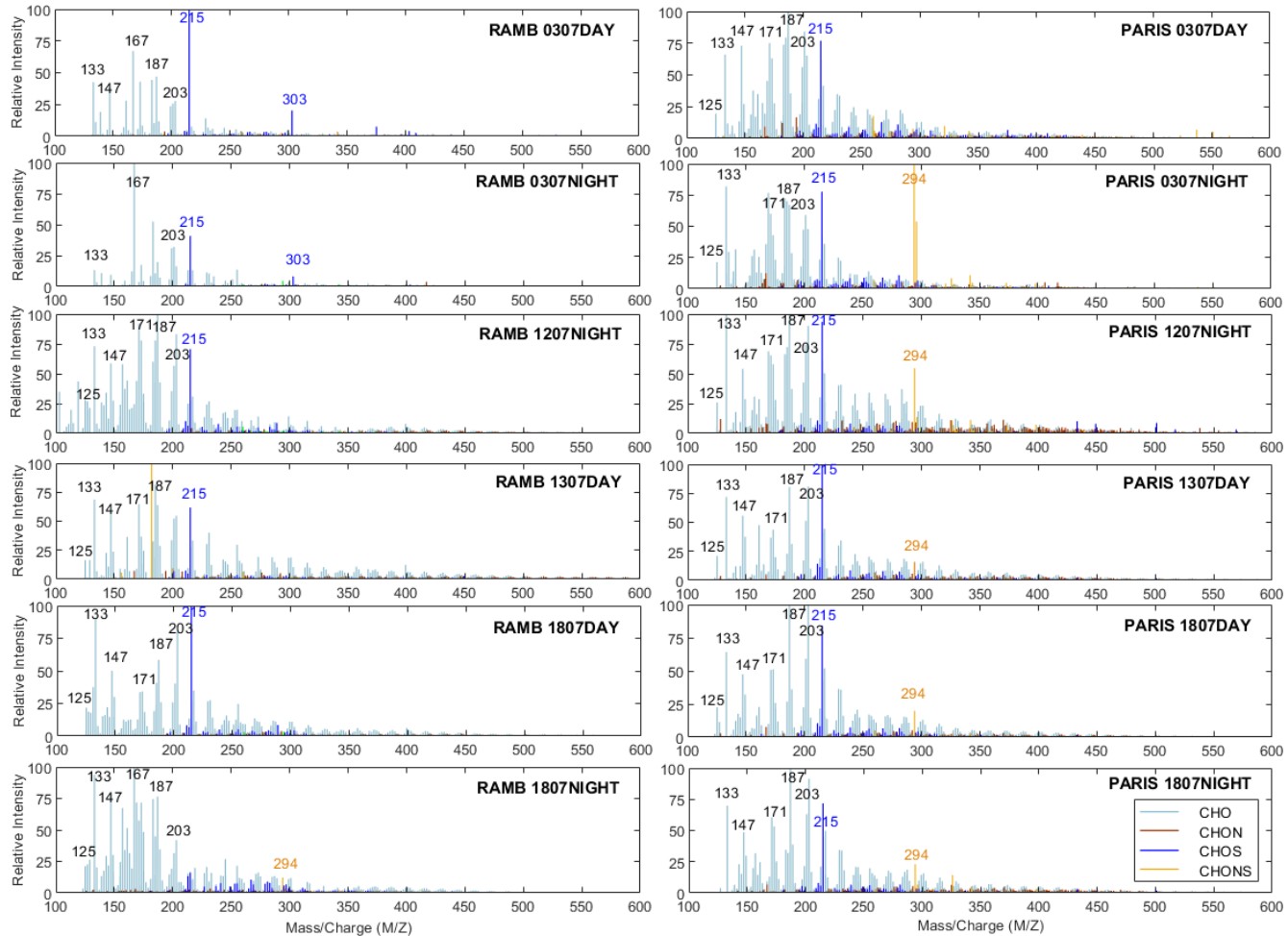

**Figure 3.** Reconstructed mass spectra for specific consecutive samples of Rambouillet (RAMB) and Paris. Different colors show the compound classes CHO (light blue), CHON (brown), CHOS (dark blue), and CHONS (orange). Samples here represent examples of the background period (3 July) and pollution periods (12, 13, and 18 July) from day and night measurements performed during summer 2022.

1,2,3-butanetricarboxylic acid (MBTCA) as oxidation products of $\alpha$-pinene ozonolysis (Kristensen et al., 2013, 2014; Yasmeen et al., 2010). Oxidized low-molecular-weight compounds such as $C_4H_6O_5$ and $C_5H_8O_5$ may be associated with carboxylic acids such as malic acid and hydroxyglutaric acid (Daellenbach et al., 2019). While malonic acid has been observed in the photo-oxidation scheme of toluene (Sato et al., 2007), hydroxyglutaric acid may be formed from monoterpene oxidation under the presence of $NO_x$ (Claeys et al., 2007; Zhang et al., 2018). The presence of $C_8H_{12}O_5$ was observed by Kourtchev et al. (2016) in the Amazon rainforest, with samples influenced by biogenic emissions and biomass burning. $C_9H_{12}O_3$ can be related to the pinene oxidation product.

Additionally, an increase in the relative intensity of $C_6H_5NO_4$ at 154.015 $m/z$ is observed on 19 July (Fig. S3) in the daytime samples at high relative intensities of 22 % and 19 % for Paris and Rambouillet, while its relative in-

tensity is lower than 4 % and 2 %, respectively, in the remaining samples. $C_6H_5NO_4$ has been previously assigned as nitrocathecols (Kourtchev et al., 2016) and attributed to biomass burning (Iinuma et al., 2010). This relative intensity variability between samples together with higher OC and $NO_x$ concentrations for that particular day supports this attribution and highlights that different chemical processes occurred during the fire event (Menut et al., 2023).

In order to provide additional information on the chemical diversity of the aerosol samples, the percentages of formulae number per compound families CHO, CHON, CHOS, and CHONS are reported in Fig. 4. The numbers of formulae associated with each group are detailed in Table S1. Samples under study were characterized by a higher number of formulae from CHO and CHON compounds (> 26 %) and lower number of CHOS and CHONS compounds. CHO and CHON showed percentages of number formulae between 30 %–45 % and 26 %–49 % in Rambouillet, while in

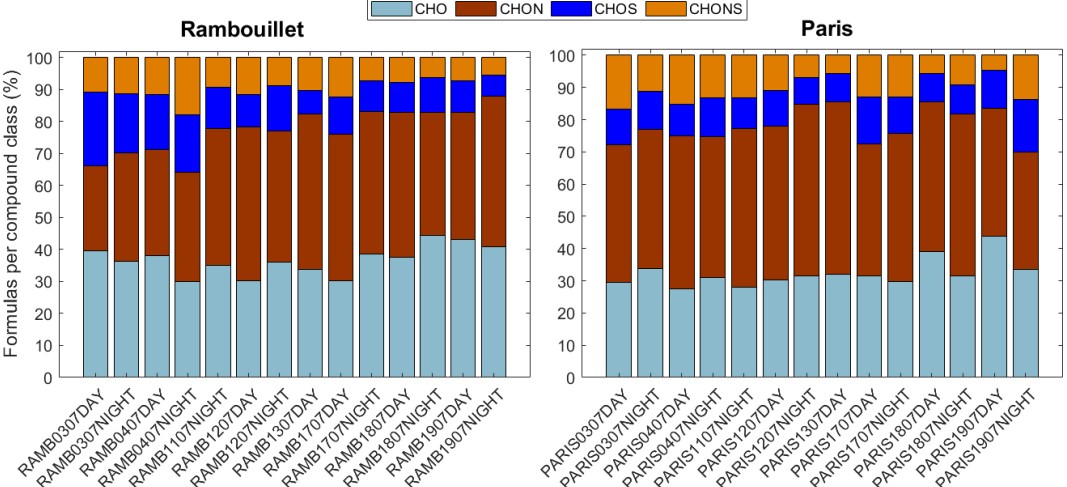

**Figure 4.** Comparison of the percentage of number of formulae per compound class for Rambouillet (RAMB) and Paris samples. Different colors show the compound classes CHO (light blue), CHON (brown), CHOS (dark blue), and CHONS (orange).

Paris they varied between 27 %–44 % and 36 %–53 %, respectively. Although molecular classes may show a seasonal variation due to the contribution of different sources (Daellenbach et al., 2019), a high abundance of CHO and, especially, CHON fractions at areas influenced by urban emissions observed here is consistent with some previous studies (Wang et al., 2018; Giorio et al., 2019; Daellenbach et al., 2019) that reported HRMS analyses of samples collected in urban environments.

Samples collected on 3 and 4 July in the forest area presented a higher percentage of number formulae of CHOS compounds ($> 17 \%$) than those collected in urban Paris ($> 10 \%$). For those days, Paris samples have a CHON presence of 43 %, which was larger than CHO ($< 34 \%$), while Rambouillet showed an opposite tendency. Besides those samples, CHON assignments either dominate or remain close to CHO assignments, except for 19 July. The presence of sulfur-containing compounds (CHOS) has shown the role of oxidizing biogenic SOA in different remote areas, which can be influenced by anthropogenic emissions such as in Manaus in the Amazon rainforest (Kourtchev et al., 2016) or in Hyytiälä in a boreal forest (Daellenbach et al., 2019).

The fact that similar percentages of molecular classes together with similar OC concentrations are observed in this work for most of the days for the periods of pollution at two locations of the Paris region may suggest similar aerosol sources influencing the chemical composition. This is not observed for the background period in the Rambouillet forest as a lower number of molecular formulae of CHON compounds were observed, as a consequence of the lower influence of anthropogenic emissions.

### Aromaticity analysis

Figure S4 shows the visual resemblance of VK diagrams for samples collected for the same days in the urban and the forested areas, with a high density of compounds with O / C < 1 and H / C < 2. Samples from Paris and Rambouillet were influenced by the presence of low-oxidation aromatic compounds and more highly oxidized molecules. Similarities in the VK for most of the samples and between the sites suggest a general consistency in aerosol sources between different days. Small differences in the density of peaks between samples were observed for 3 and 4 July; therefore, their VK diagrams in the function of the compound classes are better explored in Fig. 5.

Paris samples showed a higher density of low-oxidation aromatic compounds (region A) with main contributions of CHO, CHON, and CHONS families, while Rambouillet is shown to be less influenced by N-containing families in region A during these dates. Similarly, in region C, represented by more oxidized aerosol, the presence of CHONS compounds, especially for day samples, was abundant for the urban area. Besides those samples, both sites showed similar patterns in region A for the pollution periods. Similarly, region B (aliphatic) showed a lower density of compounds in Rambouillet during the background periods. A predominance of CHO and CHOS contributions, which can be associated with biogenic or anthropogenic first-generation products (Kourtchev et al., 2013), was observed. In Paris, an increase in the density of CHONS compounds in region B compared to Rambouillet was observed, especially for samples collected during the day.

Although there is complexity in the organic mixtures of samples collected in the field with a wide contribution of different aerosols, some patterns previously observed in the literature may suggest the compatibility of some sources

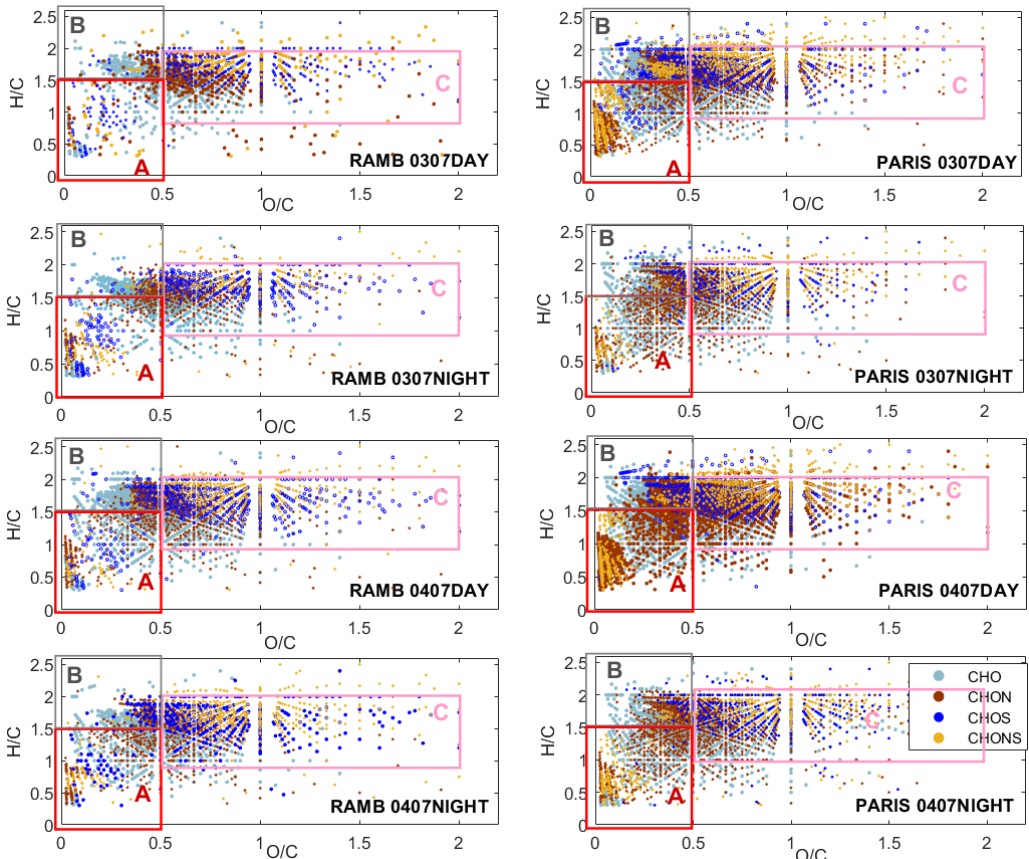

**Figure 5.** Van Krevelen diagrams for different compound classes of samples collected during day and the night periods on 3 and 4 July in Rambouillet and Paris during summer 2022. Region A represents the low-oxidation aromatic hydrocarbon domain, region B the aliphatic domain, and region C the more oxidized domain.

in Fig. 5. For example, there are possible contributions from mono- and dicarboxylic acids at the aliphatic domain and soot-derived materials or oxidized polycyclic aromatic hydrocarbons PAHs in the aromatic domain (Wozniak et al., 2008; Lin et al., 2012). The presence of lipids and fatty acids in the aliphatic domain and condensed hydrocarbons was observed in plants (Giorio et al., 2015). There was an influence on fresh and oxidized biogenic emissions (monoterpenes, limonene, isoprene, carene) for more oxidized aerosols (Kourtchev et al., 2015). Those compounds could have originated from anthropogenic sources such as vehicular emissions, which can contribute, for example, nitroaromatic compound formation. Also, contributions from biogenic sources resulting from the oxidation of terpenes and isoprene originated in forested areas with oak and pine population. Although H / C and O / C values suggested the presence of low-oxidation aromatic compounds (Koch and Dittmar, 2006; Mazzoleni et al., 2012), VK diagrams do not provide further structural information, and relying only on H / C and O / C values could be a non-accurate metric for the analysis of aromatic compounds. The aromaticity equivalents (Xc) of the aerosol samples were calculated follow-

ing Yassine et al. (2014). They were grouped as unsaturated, aromatic, and condensed aromatic compounds and reported in percentages of the total number of formulae, as shown in Fig. 6. Differences in the number contribution of unsaturated, aromatic, and condensed aromatic compounds were observed mainly at samples from the background period (3 and 4 July), where condensed aromatic compounds at Rambouillet contribute from 15 % to 22 %, while higher contributions were observed in Paris, 20 % to 28 %. An opposite trend was observed between 12 and 13 July, where condensed aromatics were more concentrated in Rambouillet than in Paris, which had slightly higher contributions of unsaturated compounds. Besides the samples of the beginning of July, strong differences in the aromaticity were not observed in terms of Xc. Comparing the variations on the percentage number of formulae for the same site, we observed an increase in the condensed fraction from the background to the pollution site for Rambouillet, although similar OC concentrations were observed at both sites. This observation highlights differences in the chemical composition at the Rambouillet site during the different periods. The average conditions highlight the lower number of formulae per aromaticity class for con-

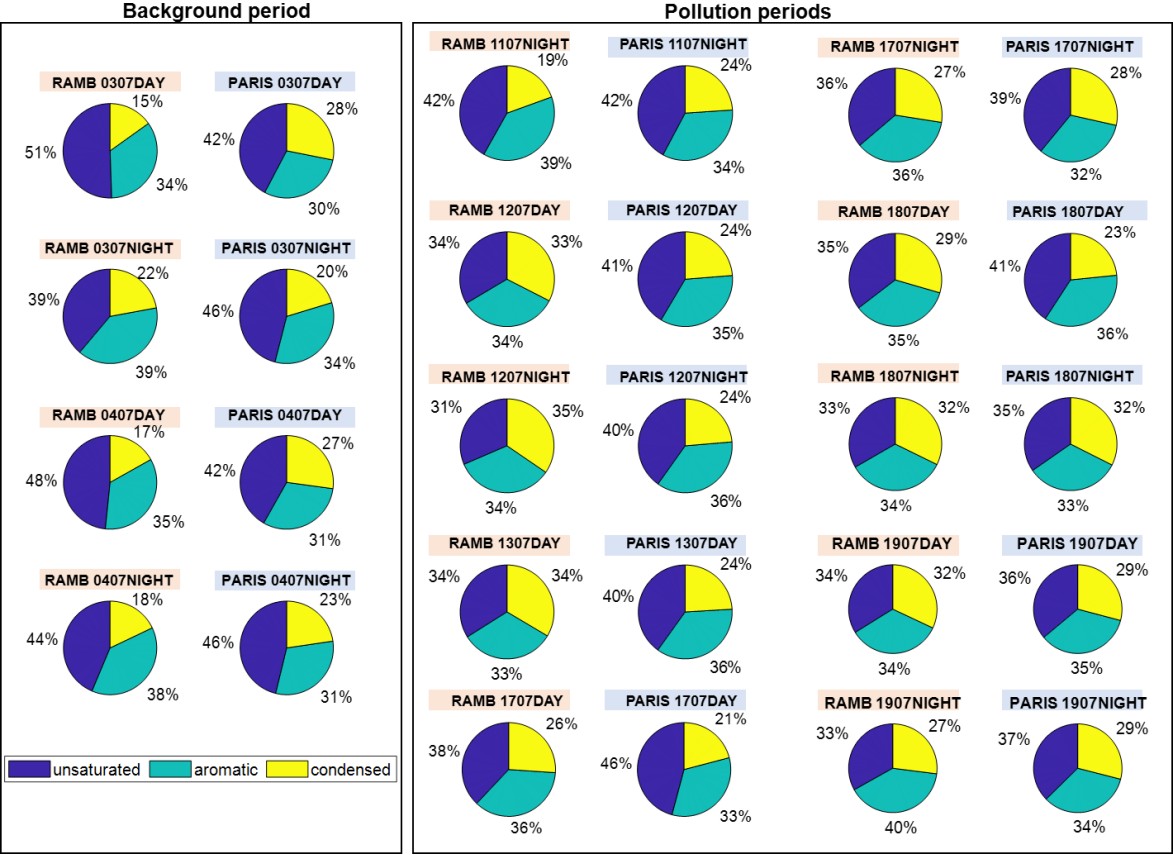

**Figure 6.** Comparison of the percentage of number of formulae per aromaticity equivalent in Rambouillet (RAMB) and Paris samples. Different colors show the compound classes unsaturated (dark blue), aromatic (turquoise), and condensed (yellow).

densed compounds, probably of aromatic origin. However, similar percentages of number of formulae were averaged for the pollution periods at both sites (Fig. S5) with average values of 40 %, 34 %, and 26 % for Paris and 35 %, 36 %, and 29 % for Rambouillet of unsaturated, aromatic, and condensed classes, respectively.

Aromatic compounds are mostly observed in areas strongly influenced by traffic emissions in the summer (Kourtchev et al., 2014). The lower number of aromatics observed in Rambouillet samples of 3 and 4 July may reflect in a better way the biogenic nature of emissions observed in a forested area under weak urban influence, also demonstrating the background signature of this period. The difference between these samples and the others is also verified by the cosine differences higher than 0.3, observed in Fig. S6, especially for samples collected during nighttime. These observations, together with the similarities on the OC concentrations and the contribution of CHON compounds observed above, may suggest air mass arrival from urban areas to this forested environment during the sampling period. A confirmation of inputs of air masses with urban/industrial contributions from Paris, Brussels, and the Ruhr region to the forested area of

Rambouillet was recently reported during the ACROSS campaign by Andersen et al. (2024).

## 3.3 Influence of chemical classes and meteorological parameters on OC concentrations

This section explores possible correlations and considers the influence of meteorological conditions observed during the campaign on the OA chemical composition. The mean values of meteorological parameters (temperature and RH) and of anthropogenic pollutant concentrations ($O_3$, $NO_x$, $SO_2$, and EC) were considered over the sampling period (Table S1) to evaluate their influence on the chemical composition of the OA through correlation analysis. The correlation coefficients ($r$) observed for Paris and Rambouillet are summarized in Fig. 7. Only statistically significant correlations (i.e., $p$ value $< 0.05$) with good and moderate linear positive/negative coefficient values are discussed ($-0.40 < r > 0.40$). Figure 7 showed good positive correlations between temperature and OC concentrations, especially for Rambouillet samples ($r = 0.59$, $p$ value $= 0.03$). The temperature may influence OC concentrations, affecting primary and secondary processes such as primary OA emissions, precursor emissions

(Sheehan and Bowman, 2001), boundary layer height, and circulation patterns (Zhang et al., 2023). These factors may affect daily variations of different compounds and in turn the OC concentration, which is observed here by the different temperature effect on the chemical families. RH was negatively correlated with $NO_x$ ($r = -0.82$, $p$ value $< 0.001$) and EC ($r = -0.75$, $p$ value $= 0.003$) in Paris. This was not the case for Rambouillet, where $NO_x$ and EC levels were lower, and instead only negative correlations with $SO_2$ ($r = -0.71$, $p$ value $= 0.006$) were observed. The lack of correlation between the meteorological conditions and pollutants in Rambouillet can be a consequence of the lack of local anthropogenic emissions in the forested site, which is supported by the low levels of EC observed. In the presence of humidity, $NO_x$ and $SO_2$ gases can transition into acidic species, decreasing their concentration and therefore being negative correlated with RH.

Positive correlations between $NO_x$ concentrations with the percentage number of molecular formulae for CHON ($r = 0.56$, $p$ value $= 0.05$) and the aromatic subgroup ($r = 0.68$, $p$ value $= 0.01$) were observed in Paris. This was not the case for the other chemical families, as $NO_x$ was negatively correlated with the percentage number of formulae for CHOS, CHONS, and unsaturated types. While positive correlations can highlight the role of $NO_x$ into the formation specific groups, the negative correlations can suggest an inhibition effect. Anthropogenic pollutants have been shown to influence the formation of organonitrate compounds (Lim et al., 2016). Also, nitroaromatic compound formation in the presence of $NO_x$ was reported by Sato et al. (2022). Possible mechanistic pathways for those groups can be derived from CHO compound oxidation, forming alkylperoxy radicals ($RO_2\cdot$) and subsequent $NO_2$ addition or NO reaction, leading to the formation of N families (e.g., organo-nitrate, peroxynitrate) (Atkinson, 2007; Kroll and Seinfeld, 2008). The correlations observed are consistent with the different roles of anthropogenic pollutants reported in the literature (McFiggans et al., 2019; Shrivastava et al., 2019).

It is important to highlight the different role of RH in the chemical families, as a positive correlation was observed with the percentage of the number formulae for aromatic compounds in Rambouillet ($r = 0.70$, $p$ value $= 0.01$) and with CHOS family in Paris ($r = 0.63$, $p$ value $= 0.02$). Differences were also observed for correlations between the percentage number of molecular formulae for different families for each site. For example, CHO was negatively correlated with CHOS ($r = -0.73$, $p$ value $< 0.007$) in Rambouillet. In Paris, aromatics were positively correlated with CHON families ($r = 0.73$, $p$ value $= 0.004$) and EC concentrations ($r = 0.65$, $p$ value $= 0.02$), which was not the case for Rambouillet. The lack of correlation between EC and aromatics for Rambouillet may suggest the depletion of aromatic compounds. These differences between sites are indicative of the different processes occurring in each environment. Negative correlations observed at both sites between the percentage

number of molecular formulae for CHO and CHOS families highlighted further oxidation processes under the presence of anthropogenic oxidants such as $SO_2$.

$O_3$ may play different roles as a photochemistry indicator as a good negative correlation with the percentage number of molecular formulae for CHOS ($r = -0.56$, $p$ value $< 0.05$), while positive correlations were observed for the percentage number of molecular formulae for CHO ($r = 0.58$, $p$ value $< 0.04$) and CHON ($r = 0.62$, $p$ value $< 0.03$) in Paris. $O_3$ is an important oxidant for biogenic precursors, but it can also correlate to other day and night oxidants such as OH and $NO_3$, respectively. The influence of $NO_x$ and $O_3$ in promoting the aerosol formation can also be observed at the molecular scale in Fig. S7, for common formulae (577) from different families of compounds in both Paris and Rambouillet. In Fig. S7, $NO_x$ concentration is positively correlated with the percentage of number formulae for compounds in the low-oxidation aromatic domain, while $O_3$ influences the more oxidized region, which also seems positively correlated with the OC concentrations, highlighting the importance of the secondary contribution to OA formation. The correlations observed suggest a potential link between the meteorological parameters, anthropogenic pollutants, and the OA composition; however, further research is needed to fully understand the extent of this impact.

## 4 Comparison of the chemical composition

This work aimed to provide a description of chemical composition of the organic fraction of the aerosol to investigate the differences in $PM_1$ collected in the urban and forested areas of Paris from day and night measurements during summer 2022. Lower values of OC together with lower $NO_x$ concentrations observed during the background period highlighted an atmosphere less influenced by anthropogenic inputs in the forested area. For the defined polluted periods, similar aerosol chemical compositions were observed for both urban and forested areas with OC concentration values in agreement with previous studies carried out in the Paris region (Bressi et al., 2013; Lanzafame et al., 2021).

The presence of $NO_x$, CHON and CHONS species, and aromatic and condensed aromatic compounds detected in Rambouillet samples together with the air mass back-trajectories previously reported highlights the impact of urban inputs in forested areas. Similarly, the important density of peaks, probably associated with biogenic contributions identified in the Paris center samples (e.g., $C_8H_{12}O_4$, $C_8H_{12}O_6$, $C_{10}H_{17}NSO_7$), should also be noted. As wind directions from the northeast and west were observed for Paris during the sampling period, the detection of biogenic compounds may be influenced either by biogenic VOC emissions from natural areas close to the sampling site (e.g., Vincennes and Boulogne forests or urban trees) or by direct biogenic aerosol inputs. This observation is consistent with previous

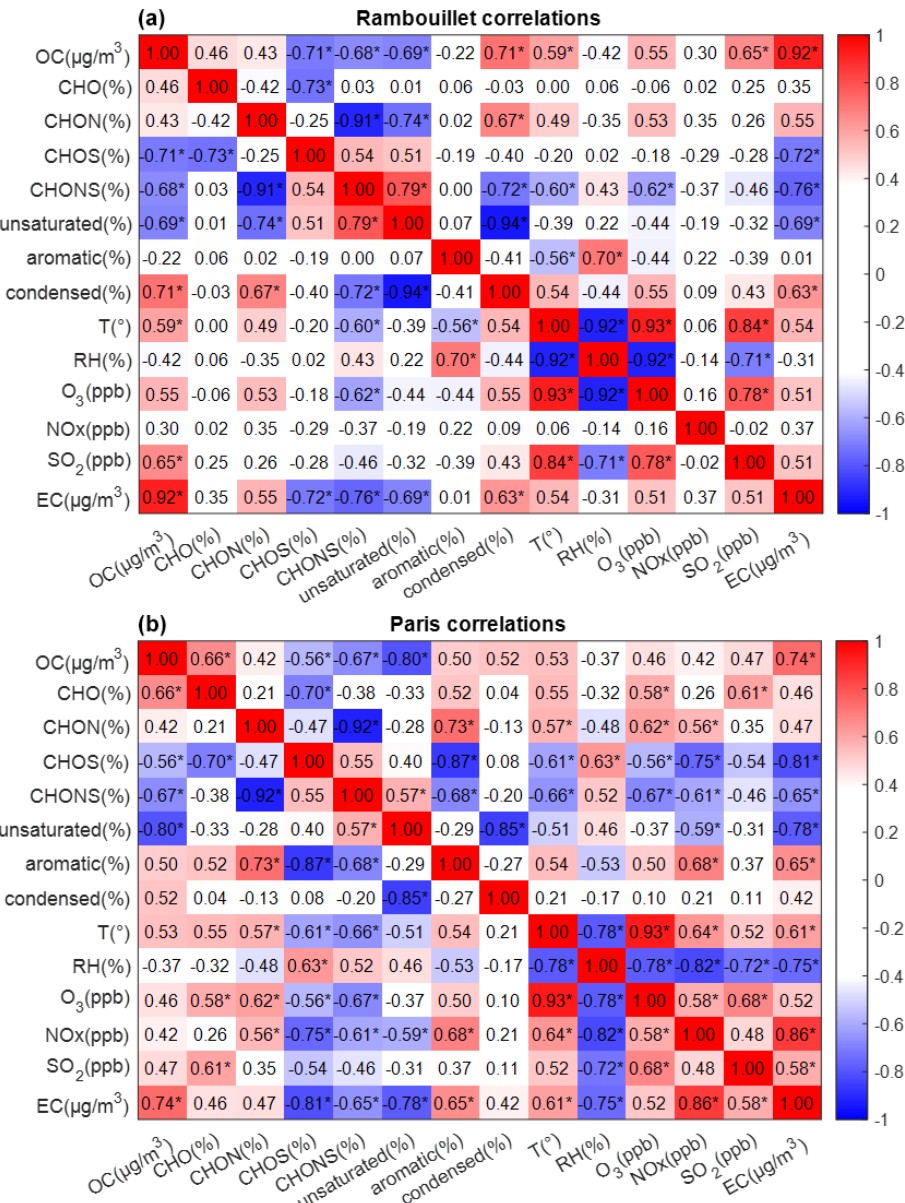

**Figure 7.** Correlation coefficient matrix for compound classes (CHO, CHON, CHOS, CHONS, and CHONS), meteorological conditions (temperature and RH), anthropogenic pollutants ($O_3$, $NO_x$, $SO_2$, and EC), and OC concentrations observed for Rambouillet **(a)** and Paris **(b)**. The Pearson correlation coefficients for negative correlation (blue) and positive correlations (red) are presented here. * shows statistically significant ($p$ value $< 0.05$) correlation values.

studies showing that biogenic compounds can play an important role during the summer in urban areas (Amarandei et al., 2023; Giorio et al., 2019; Maison et al., 2024).

The molecular classes (CHO, CHON, CHONS, and CHOS) identified in this work may originate from both biogenic and anthropogenic sources, influencing both background and the pollution periods. The different compound classes show a predominance of CHO ($> 27\%$) and CHON ($> 26\%$) groups, consistent with reported contributions in the literature: 25% and 48% in Padua (Italy) (Giorio et al., 2019), 45% and 21% in Iași (Romania) (Amarandei et al., 2023), 44% and 21% in Beijing (China) (Wang et al., 2018), and 32% and 35% in Mainz (Germany) (Wang et al.,

2018). These similarities suggest the ubiquitous contribution of some aerosol component both of anthropogenic and biogenic origin in urban areas.

Although no significant difference was observed for OC concentrations between daytime and nighttime, the higher relative intensity of isomeric compounds such as $C_{10}H_{17}NSO_7$ (more prominent in the Paris area) highlights the different processes and sources and potential variations of species concentrations during the day. The description of the OA chemical composition at molecular scale improves the understanding between the mixing of biogenic and anthropogenic emissions as, for example, organosulfur compounds such as $C_5H_{12}SO_7$ and $C_{10}H_{17}NSO_7$ were detected. They have been previously observed in environments influenced by urban emissions (Kourtchev et al., 2014; Kourtchev et al., 2016; Giorio et al., 2019; Wang et al., 2022), showing the impact of mixed anthropogenic–biogenic air masses. Interactions between different biogenic and anthropogenic components (Rattanavaraha et al., 2016; McFiggans et al., 2019; Shrivastava et al., 2019) were previously reported to influence the OA composition and formation efficiency. Additionally, although only direct infusion analysis was performed in this work, the detection of molecular formulae compatible with those of the literature for possible biogenic (e.g., $C_8H_{12}O_4$, and $C_8H_{12}O_6$) (Kristensen et al., 2013, 2014) and anthropogenic compounds (e.g., $C_6H_5NO_4$) (Iinuma et al., 2010) highlights the importance of providing the chemical composition at molecular scale as organic tracers can be detected. Those can be later associated with specific SOA precursors and formation pathways. As observed, molecular-scale information provides fingerprints on the organics composition in the Paris megacity and a forested area.

Similarities found in the particle chemical composition on the samples under study derived from HRMS analysis and the temporal series of OC and $O_3$ concentrations for the pollution periods highlight a homogeneity (source consistency) in the OA composition for both urban and forested areas of Paris when anthropogenic emissions increase, consistent with previous observations of aerosol composition for urban, suburban, and rural Parisian areas (Bressi et al., 2013). These observations show that forested areas can be affected by anthropogenic inputs, influencing the atmospheric chemical composition and therefore their impact on the OA budget and related processes.

## 5 Conclusions

Aerosol filter sampling was performed during the ACROSS intensive campaign at two sites in the greater Paris area during summer 2022 to investigate the chemical composition of the organic fraction of $PM_1$ at the molecular scale at two sites representative of urban (Paris) and forested (Rambouillet) environments. The OC concentrations derived in this work were similar for both sampling sites and in agreement with

values previously reported for the Paris region, suggesting the influence of the urban inputs on the suburban forested area of Rambouillet. HRMS analysis showed similar patterns of the contributions of anthropogenic and biogenic emissions at both sites for periods of pollution. This was not the case for samples of the background period on 3 and 4 July, more representative of the local emissions at both sites, i.e., highlighting biogenic contributions in Rambouillet and anthropogenic sources in Paris. This observation was confirmed by statistical analysis, which showed the influence of different process occurring at both sites, together with the aromaticity analysis, which shows a higher presence of condensed aromatic compounds in Paris than in Rambouillet, with a higher density of peak assignments in the VK diagrams. The high number contribution of CHO and CHON families in both sites verified the aerosol source homogeneity for periods of pollution. Similar OA composition observed at the urban and forested areas of the Paris region during such periods reflects anthropogenic and biogenic emissions interactions enhancing summertime aerosol formation. This observation depicts a progressive impact of densely populated areas (megacities) on rural/forested areas with increasing emissions. Additionally, the detection of tracers such as $C_5H_{12}SO_7$, $C_{10}H_{17}NSO_7$, and $C_6H_5NO_4$ observed at both sites completes this statement, showing the contribution of mixed biogenic emissions and biomass-burning sources, and highlights the importance of using molecular tracers in the description and quantification of the organic fraction of the aerosol. These observations provide the first HRMS molecular screening analysis for the Paris region, improving the understanding of OA composition and the differences and similarities between urban and forested areas. However, further information is needed to compare the chemical composition in different environments in a quantitative way to properly assess the mixing between air masses and their global impact on modeling and air quality studies.

**Data availability.** Data presented in this work for the two sampling sites are available at the AERIS (French national center for atmospheric data and services) facility (https://across.aeris-data.fr/catalogue/, last access: 7 September 2024). The data sets are available for OC and EC concentrations for Paris (https://doi.org/10.25326/682, Pereira et al., 2024a) and Rambouillet (https://doi.org/10.25326/681, Pereira et al., 2024c), the HRMS analysis for Paris (https://doi.org/10.25326/683, Pereira et al., 2024b) and Rambouillet (https://doi.org/10.25326/684, Pereira et al., 2024d), and meteorological information for Paris (https://doi.org/10.25326/573, Di Antonio et al., 2023) and Rambouillet (https://doi.org/10.25326/437, Denjean, 2023).

**Supplement.** The supplement related to this article is available online at https://doi.org/10.5194/acp-25-1-2025-supplement. TS5

**Author contributions.** DLP, AG, CG, and PF designed the research. PF, VM, CC, CD, AG, DLP, MC, GN, SC, SA, EA, AB, TB, MC, PC, LDA, SH, JH, CG, OG, BL, OL, CM, FM, BPV, RT, ST, PZ, LW, DP, SR, PMF, EP, PP, EV, AA, OF, RAP, and JFD participated in sample collection and/or instrument deployment in the field. DLP, CG, GN, and AZ conducted the filter analysis. DLP, CG, AZ, AG, and PF analyzed the data. DLP drafted the initial manuscript. CG, AZ, AG, PF, PP, EP, JFD, and EV reviewed and corrected the manuscript. CG and AZ provided the expertise on the HRMS analysis. CC and VC are the principal investigators of the ACROSS project. All authors made contributions to this work and approved the final version of the manuscript.

**Competing interests.** At least one of the (co-)authors is a guest member of the editorial board of *Atmospheric Chemistry and Physics* for the special issue "Atmospheric Chemistry of the Suburban Forest – multiplatform observational campaign of the chemistry and physics of mixed urban and biogenic emissions". The peer-review process was guided by an independent editor, and the authors also have no other competing interests to declare.

**Disclaimer.** Publisher's note: Copernicus Publications remains neutral with regard to jurisdictional claims made in the text, published maps, institutional affiliations, or any other geographical representation in this paper. While Copernicus Publications makes every effort to include appropriate place names, the final responsibility lies with the authors.

**Special issue statement.** This article is part of the special issue "Atmospheric Chemistry of the Suburban Forest – multiplatform observational campaign of the chemistry and physics of mixed urban and biogenic emissions". It is not associated with a conference.

**Acknowledgements.** The ACROSS and the PEGASUS databases and their access are maintained by the French national center for atmospheric data and services (AERIS) as part of the French research infrastructure DataTerra. The authors acknowledge the help of Martin des Forges and Marion Cayet during the filter preparation.

**Financial support.** This research has been supported by the French National Research Agency (ANR, grant no. ANR-17-MPGA-0002), the French national program LEFE of CNRS-INSU, the project TRAC-AOS-A within the LEFE-CHAT national program from CNRS-INSU and from ADEME, a BP Next Generation Fellowship awarded by the Yusuf Hamied Department of Chemistry at the University of Cambridge, the IDEX program of the Université Paris Cité, and the French Ministry of Environment.

**Review statement.** This paper was edited by Alexander Laskin and reviewed by two anonymous referees.

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

## Remarks from the typesetter