# Peer review of "Molecular characterization of organic aerosols in urban and forested areas of Paris using high resolution mass spectrometry"

_EGUsphere, 2024_

## Referee Comment (RC1)

**General Comments:**

This manuscript (MS-ID: egusphere-2024-3015) presents a comprehensive study on the molecular characterization of organic aerosols in urban and forested areas of the Paris region using high-resolution mass spectrometry (HRMS). The authors provide valuable insights into the chemical composition, sources, and potential interactions between anthropogenic and biogenic emissions that influence the organic aerosol characteristics in this mixed environment. However, the current version of manuscript should be strengthened by providing more detailed explanations and discussions on the observed patterns, the potential drivers of the urban-rural differences, and the implications of the findings for regional air quality and atmospheric chemistry. Additionally, the authors need to consider addressing the specific comments and technical suggestions provided to improve the overall clarity, coherence, and presentation of the manuscript. Overall, this manuscript can be published after addressing these major concerns and issue listed below. But, I would let editor to decide.

**Major Concerns:**

1. On page 3, lines 65-70, the authors claim that the Paris area is surrounded by low urbanized areas mostly composed of intensive agriculture fields and forest. But it is not clear how this spatial distribution of urban and rural areas may influence the mixing of anthropogenic and biogenic emissions. The authors should provide more details on the potential transport and interactions between these different air masses. (Page 3, lines 65-70)
2. Line 150, the authors applied methanol to extract filter sample, what about other solvents like water or $CH_3CN$. How can authors ensure that methanol can extract all major OAs in the filter samples. Additional experimental evidence needs to be supported.
3. What's the extraction efficiency of methanol?
4. Line 168, The authors state that CHN and CHS families were not considered here due to limitation of test mode. But I would like to know the abundance of these compounds and how they compete with CHO, CHON, CHOS, CHONS, and CHNS?
5. Line 180, what's the uncertainties or biases for the assumption of using carboxylic acids (R-COOH) instead of other likely compounds? This would eventually have the impact on the classification of aromatics types.
6. The authors mention the detection of C5H12SO7 and C10H17NSO7 compounds, which they attribute to isoprene and α-pinene oxidation products, respectively. The authors should provide more details on the potential formation mechanisms and the role of anthropogenic pollutants (NOx and SO2) in the formation of these organosulfate compounds.

Otherwise, it would be great challenge to allow readers to follow the interpretation. (Page 11, lines 277-283)

7. In section 3.2.1, the authors discuss the similarities and differences in the Van Krevelen diagrams between the urban and forested sites. While they provide some interpretations of the different compound domains, more detailed explanations are needed on the potential sources and formation processes associated with the observed patterns. (Page 14, section 3.2.1)

8. In section 3.1, the authors discuss the temporal variability of OC and EC concentrations at the two sites. However, they do not provide a clear explanation for the observed similarities and differences between the urban and forested sites. More discussion is needed on the potential drivers of these patterns, such as the influence of meteorology, air mass transport, and local versus regional sources. (Page 7, section 3.1)

9. The authors mention the detection of $C_5H_{12}SO_7$ and $C_{10}H_{17}NSO_7$ compounds, which they attribute to isoprene and α-pinene oxidation products, respectively. More details on the potential formation mechanisms and the role of anthropogenic pollutants ($NO_x$ and $SO_2$) in the formation of these organosulfate compounds are required. (Page 10, lines 277-283)

10. The authors perform a correlation analysis to investigate the influence of meteorological parameters and anthropogenic pollutants on the chemical composition of the organic aerosols. The discussion of these results could be strengthened by providing more mechanistic explanations for the observed correlations, particularly for the differences between the urban and forested sites. (Page 16, section 3.3)

11. The authors conclude that the forested area of Rambouillet is affected by anthropogenic inputs, influencing the atmospheric chemical composition. A clear quantification or assessment of the relative contributions of anthropogenic versus biogenic sources to the organic aerosol composition at the two sites would help to better understand the extent of the urban influence on the forested area. (Page 19, lines 464-467)

12. The authors state that the HRMS analysis showed similar patterns of the contributions of anthropogenic and biogenic emissions on both sites for periods of pollution. However, a clear definition or criteria for what constitutes a "period of pollution" is missing. More information is needed on how these periods were identified and how they differ from the background conditions. (Page 19, lines 469-471)

13. The similarities in the chemical composition between the urban and forested sites during the pollution periods have been discussed and the authors are encouraged to supply the potential implications of these findings for the regional air quality and the impact of urban emissions on the surrounding environment. (Page 19, lines 469-471)

14. The authors state that the detection of compounds associated with biogenic and anthropogenic oxidation products highlights the importance of understanding urban and rural chemistries at the molecular level. However,

they do not provide a clear discussion on how this molecular-level information can be used to improve our understanding of the complex interactions between different emission sources and their impact on the regional atmospheric composition. (Page 19, lines 449-457)
15. The authors mention the use of the ACROSS campaign data, but they do not provide any details on the specific objectives, experimental design, or other relevant information about this campaign. (Page 4, lines 110-115)
16. A big picture or schematic is strongly recommended to summarize the finding in this work.

Technique Issue:

1. In Figure 2, the y-axis labels for the mass spectra are not clearly legible. Please consider increasing the font size or adjusting the layout to improve the readability of the figure.
2. Figure S3 are not clearly legible and unreadable.

---

## Author Comment (AC1)

We are thankful for the revisions on the manuscript provided by the reviewers. Please, find our responses to each point below.

**Reviewer 1**
General Comments:

This manuscript (MS-ID: egusphere-2024-3015) presents a comprehensive study on the molecular characterization of organic aerosols in urban and forested areas of the Paris region using high-resolution mass spectrometry (HRMS). The authors provide valuable insights into the chemical composition, sources, and potential interactions between anthropogenic and biogenic emissions that influence the organic aerosol characteristics in this mixed environment. However, the current version of manuscript should be strengthened by providing more detailed explanations and discussions on the observed patterns, the potential drivers of the urban-rural differences, and the implications of the findings for regional air quality and atmospheric chemistry. Additionally, the authors need to consider addressing the specific comments and technical suggestions provided to improve the overall clarity, coherence, and presentation of the manuscript. Overall, this manuscript can be published after addressing these major concerns and issue listed below. But, I would let editor to decide.

The authors thank to the reviewer for the careful revisions and comments that help us to improve the overall discussion of this manuscript. Regarding the specific comments and technical suggestions, they have been addressed below to expand the discussion of the chemical composition of the organic aerosol.

Major Concerns:

1. On page 3, lines 65-70, the authors claim that the Paris area is surrounded by low urbanized areas mostly composed of intensive agriculture fields and forest. But it is not clear how this spatial distribution of urban and rural areas may influence the mixing of anthropogenic and biogenic emissions. The authors should provide more details on the potential transport and interactions between these different air masses. (Page 3, lines 65-70)

Parisian areas influenced by human activities can contribute with anthropogenic emissions. In contrast, dense vegetated areas may contribute mainly with biogenic emissions. The fact that the urban area of Paris is surrounding by suburban and rural areas provides a wide contribution of different aerosol and gaseous species of both origins. Due to atmospheric dynamics, those emissions can encounter during the air masses transport. Certain atmospheric conditions, such as anticyclone systems can favour these air masses interactions due to the slow moving conditions (Lagmiri & Dahech, 2023; Wei et al., 2011). The impact of anticyclone conditions has previously led to the identification of PM accumulation (Beekmann et al., 2015) and variability of aerosol particles (Bressi et al., 2013) at the Paris region. In this region, wind roses from measurements previously performed in the summer have shown the important contribution of air masses arriving not only from the highly urbanized areas (northeast), but from air masses traveling into west directions, crossing the urban area of Paris and going through low urbanized and forested areas (https://across.cnrs.fr/across-white-position-paper/).

*Lines 70 to 74 were rewritten as follows to clarify this point:*
*Mixing between anthropogenic and biogenic emissions can occur, especially when the plume of Paris travels away from the city. This encounter can be favour by anticyclonic weather that allows the interaction between air masses due to slow moving conditions (Lagmiri & Dahech, 2023; Wei et al., 2011). Anticyclone*

*conditions has previously led to the identification of PM accumulation (Beekmann et al., 2015) and variability of aerosol particles (Bressi et al., 2013) at the Paris region.*

**2. Line 150, the authors applied methanol to extract filter sample, what about other solvents like water or CH₃CN. How can authors ensure that methanol can extract all major OAs in the filter samples. Additional experimental evidence needs to be supported.**

Methanol has been widely used in Orbitrap analysis due to its general suitability for OA (Giorio et al., 2019; King et al., 2019; Kourtchev et al., 2013, 2014, 2016; Tong et al., 2016) collected at field and chamber studies for biogenic and anthropogenic aerosol. We agree with the reviewer that molecular composition is affected by the choice of the solvent and its polarity. For example, recently we have shown that methanol extracts are enriched in unsaturated components as compared to water extraction from urban OA (Zherebker et al., 2024). But for direct infusion analysis water extracts should be desalted, which leads to the carbon loss due to selectivity of the resin for solid-phase extraction. Therefore, for our study we selected organic solvent. It allowed to extract polar species ionizable by negative electrospray and to avoid extra purification step. Previous comparison of acetonitrile (LC-MS grade) and methanol (LC-MS grade) demonstrated a lack of differences in molecular composition of extracts (Kourtchev et al., 2013). Methanol is cost effective. Considering these parameters, we used methanol extraction consistently for convenience and proven efficiency for OA. We have added the following lines on the text:

*Lines 171 to 173: Methanol was used due to its suitability for the extraction of polar unsaturated compounds (Zherebker et al., 2024) and high extraction efficiency of OA (Giorio et al., 2019) enabling analysis without extra purification step.*

**3. What's the extraction efficiency of methanol?**

In this research, the protocol used in Kourtchev et al., 2014 have been applied and quantitative measurements of carbon recovery and extraction efficiency were not performed. However, methanol has proven to be a suitable solvent with high extraction efficiency (Chen & Bond, 2010; Cheng et al., 2016; Mihara & Mochida, 2011; Xu et al., 2022; Yan et al., 2020) for polar organic constituents providing comparable matrices for analysis. Other organic solvents, such as dichloromethane, tetrahydrofuran, ethyl acetate, methanol, water and their mixtures could influence the extraction efficiency of certain types of organic aerosol. However, as the same extraction procedure was applied for all the samples in this work, it is a reasonable to compare mass-spectra of extracts without considering carbon concentration.

**4. Line 168, The authors state that CHN and CHS families were not considered here due to limitation of test mode. But I would like to know the abundance of these compounds and how they compete with CHO, CHON, CHOS, CHONS, and CHNS?**

Thank you for the question. We agree with the reviewer that the other families can also contribute to the OA chemical composition. Therefore, the contribution of all the possible families (CHO, CHON, CHONS, CHOS, CHNS, CHN, CHS) is shown in Figure R1 for Paris and Rambouillet. As observed the families CHS, CHN and CHNS accounts for less than the 1%, 5%, and 2% of the family's contribution for both sites. The poor ionization efficiency of those families impacts their detection and resolution, preventing reliable formula assignment. Therefore, we considered those families don't influence the overall chemical composition, and therefore, we focus on O-containing compounds (CHO, CHON, CHONS and CHOS) due to the higher ionization efficiencies. We have correct the text as follows and modify table 2, to remove the CHNS contribution. CHNS contribution from figures 3 and 5 were also removed.

*Lines 193-197: The formulae assigned were grouped based on the molecular composition as CHO, CHON, CHOS, CHONS, CHNS, CHN and CHS families. However, due to the higher ionization efficiency in the negative mode, this work focus on O-containing compounds (CHO, CHON, CHONS and CHOS). This selection is also supported by the predominance of oxygenated compounds on the OA on the PM₁ fraction reported at different urban and rural areas* (Q. Zhang et al., 2007)

.

[Figure]

*Figure R1.Percentage contribution of the CHO, CHON, CHONS, CHOS, CHNS, CHN, CHS families observed at Rambouillet (upper panel) and Paris samples (bottom panel).*

5. Line 180, what's the uncertainties or biases for the assumption of using carboxylic acids (R-COOH) instead of other likely compounds? This would eventually have the impact on the classification of aromatics types.

The aromaticity equivalent (Xc) is a parameter that can be used to sort compounds based on the carbon backbone structure by assuming main functionalities defined by the fractions m and n. The latest designates the fractions of O and S atoms involved in the $\pi$-bonds. m,n values of 0 have associated to alcohol, ether and peroxide functions, values of 0.5 were mainly associated to carboxylic acid, esters, and nitro classes, while values of 1 can be related to aldehyde, ketones, nitroso and cyanate classes (Yassine et al., 2014). Given that orbitrap analysis discussed in this study were performed in the negative ionization mode, detection of acidic species is favored, therefore we selected values of 0.5.  m=0.5 indicates that two oxygen atoms contribute to the $\pi$-bond in the carboxylic acid functionality, while n=0.5 shows that sulphur is

participating in two double bounds with oxygen atoms, such as in sulfates groups. Additionally, major contributions (>30%) of compounds containing carboxylic acid functionalities have been observed in the aerosol chemical composition (Horník et al., 2020; Michoud et al., 2021), which supports this selection.

We have evaluated the impact of assuming other functionalities for a sample of July 12 collected during the day at Paris (Figure R2). Differences are observed for the condensed aromatic contribution and unsaturated at comparing m,n=0 and m,n=0.5 as a decrease on condensed aromatic compounds from 34% to 25% is observed at increasing the value. At comparing m,n=1, an impact is observed mainly for the unsaturated fractions varying from 41% to 52%.

[Figure]

*Figure R2. Aromaticity analysis performed in one of the samples collected at the Paris urban site (PARIS1207 DAY) considering different heteroatoms fractions (m, n). Values of 0.5 (left panel) represents carboxylic acid, esters, and nitro functions. Values of 0 (middle panel) represent alcohol, ether and peroxide functions, and values of 1 (right panel) represent aldehyde, ketones, nitroso and cyanate functions.*

*We have modified the text in the manuscript to clarify this point:*
*Lines 207 to 2015: The $m,n$ parameters designates the fractions of O and S atoms involved in the π-bonds. In order to provide a conservative estimate for unsaturation accounted for only carbon-carbon π-bonds, we used m,n = 0.5, which correspond to the maximum possible fraction of $sp^2$-hybridized oxygen atoms as carboxyl groups. While conservative approach provides only a lower boundary for the contribution of*

*highly unsaturated compounds (tentatively assigned as aromatic and condensed compounds), a predominance of carboxylic acid species over other functionalities is expected for analysis in negative ion mode (Kourtchev et al., 2016). The role of carboxylic acids as one of the major fractions of aerosol particles was observed during the summer for field samples at European cities at Corsica (Michoud et al., 2021) and Czech Republic (Horník et al., 2020). Although this assumption can lead to an underestimation of possible aromatic fraction, we considered the m,n=0.5 as previously applied in the literature (Tong et al., 2016; Yassine et al., 2014).*

6. The authors mention the detection of C5H12SO7 and C10H17NSO7 compounds, which they attribute to isoprene and α-pinene oxidation products, respectively. The authors should provide more details on the potential formation mechanisms and the role of anthropogenic pollutants (NOx and SO2) in the formation of these organosulfate compounds. Otherwise, it would be great challenge to allow readers to follow the interpretation. (Page 11, lines 277-283)

As suggested by the reviewer, we have included the description of organosulfate formations as described in the literature. The following text was added to the main manuscript:

*Lines 347 to 353: The dependence on organosulfate formation in the presence of anthropogenic species has been followed during the oxidation of monoterpenes and isoprene. The presence of acidic sulfate seeds (Iinuma et al., 2009; Surratt et al., 2007) and additional presence of NOx species for the nitroxy types (Surratt et al., 2008) were observed. The mechanisms proposed for organosulfates formation involves sulfate species reactions via epoxide formation, esterification, aqueous reactions or direct reaction with gaseous $SO_2$ (Brüggemann et al., 2020). While nitrate radical chemistry was suggested to determine nitroxy organosulfates formation (Iinuma et al., 2007).*

7. In section 3.2.1, the authors discuss the similarities and differences in the Van Krevelen diagrams between the urban and forested sites. While they provide some interpretations of the different compound domains, more detailed explanations are needed on the potential sources and formation processes associated with the observed patterns. (Page 14, section 3.2.1)

Thank you for the question. Van Krevelen diagrams is a visualization of formulae assignments projected on H/C vs O/C 2D plot. This diagram can then be used to highlight molecular components based on tentative classes such as low oxygenated aromatic hydrocarbons, aliphatic and more oxidized (Kourtchev et al., 2016; Nozière et al., 2015). While such fingerprinting is a powerful tool for visual comparison of molecular composition, mass spectrometry analysis without tandem experiments and visualization techniques don't provide structural information required for a description of chemical processes in OA. Recently we have demonstrated that chemical series derived from molecular formulae (e.g. difference in the number of oxygen atoms) don't correspond to the actual transformations (Zherebker et al., 2024). Nevertheless, changes in molecular composition are being picked up from van Krevelen diagrams as well as tentative chemical classes of determined ions. More detailed source apportionment or structural analysis of SOA components was out of scope of the presented research.

*The following lines were added into the manuscript:*
*Line 427-431: Besides those samples, both sites showed similar patterns at region A for the pollution periods. Similarly, region B (aliphatic) showed a lower density of compounds at Rambouillet during the*

*background periods. A predominance of CHO and CHOS contributions, which can be associated to biogenic or anthropogenic first generation products (Kourtchev et al., 2013), was observed. At Paris, an increase on the density of CHONS compounds in region B compare to Rambouillet was observed, especially for samples collected during the day.*

*Lines 439-441: Those compounds could be originated from anthropogenic sources such as vehicular emissions, which can contribute for example, with nitroaromatic compounds formation. Also, contributions of biogenic sources resulting from the oxidation of terpenes and isoprene originated in forested areas with oaks and pines population.*

8. In section 3.1, the authors discuss the temporal variability of OC and EC concentrations at the two sites. However, they do not provide a clear explanation for the observed similarities and differences between the urban and forested sites. More discussion is needed on the potential drivers of these patterns, such as the influence of meteorology, air mass transport, and local versus regional sources. (Page 7, section 3.1)

Thank you for the comment. We have expanded the discussion around OC and EC concentrations to better understand the comparison between both sites. The following text was added to the main manuscript:

*Lines 253- 262: The difference between the forested and urban environment is observed at looking into the EC concentrations, which are higher in Paris due to the densely populated area, which translates into higher local anthropogenic activities. In contrast, similarities between the profiles of the OC concentrations for the common period between (June 27 to July 22) both sites may suggest common organic sources influencing the chemical composition. Given that different wind directions are observed after July 8, these similarities can be influenced by local sources and transported air masses. Previous summer measurements (Gros et al., 2007) of OC in the Paris urban area highlight local traffic contributions. As observed at both sites, the increasing of the OC concentrations at the end of July was accompanied by an increase of temperature, NOx and O₃, which can enhance the chemistry and derive into a higher OA formation (Luo et al., 2021; Shrivastava et al., 2019; Xu et al., 2022). Further influence of the meteorological parameters and anthropogenic pollutants into the OC concentrations and composition are explored in Section 3.3.*

9. The authors mention the detection of C5H12SO7 and C10H17NSO7 compounds, which they attribute to isoprene and α-pinene oxidation products, respectively. More details on the potential formation mechanisms and the role of anthropogenic pollutants (NOx and SO2) in the formation of these organosulfate compounds are required. (Page 10, lines 277-283)

As suggested by the reviewer, we have included the description of organosulfate formations as described in the literature (See answer question 6).

10. The authors perform a correlation analysis to investigate the influence of meteorological parameters and anthropogenic pollutants on the chemical composition of the organic aerosols. The discussion of these results could be strengthened by providing more mechanistic explanations for the observed correlations, particularly for the differences between the urban and forested sites. (Page 16, section 3.3)

As suggested by the reviewer, we have extended our explanations for the observed correlations. Addtionally, the correlation plot and values were updated at excluding the CHNS contributions. The following text was added into the main manuscript:

*Lines 488-490: The lack of correlation between the meteorological conditions and pollutants in Rambouillet can be a consequence of the lack of local anthropogenic emissions in the forested site, which is supported by the low levels of EC observed.*

*Lines 494-504: CHO compounds can undergo O₂ oxidation, forming alkylperoxy radicals (RO₂·), and subsequent NO₂ addition or NO reaction, leading to the formation of N-families (e.g., organo-nitrate, peroxynitrate)* (Atkinson, 2007; Kroll & Seinfeld, 2008). *Therefore, in urban environments, RO₂· reactions with NOₓ represent an important pathway for SOA formation. The fact that NOₓ and CHON are positively correlated only in Paris highlights the formation of those compounds on urban environments. Following this oxidation scheme, aromatic compounds are able to form nitro-aromatic compounds (Sato et al., 2022), showing also a good correlation between the aromatic subgroup and NOₓ in Paris ((r= 0.68, p-value=0.01). The fact that the chemical family's correlations with the anthropogenic pollutants follow different tendencies for both sites may suggest that some species are formed in the urban area and then transported to the forested site. NOₓ was also negative correlated with the percentage number of formulae for CHOS, CHONS, and unsaturated types, showing they those compounds can react in the presence of this anthropogenic pollutant, known as a major oxidant, especially during the night* (Atkinson & Arey, 2003).

[Figure]

**Figure 1: Correlation coefficient matrix for compounds classes (CHO, CHON, CHOS, CHONS and CHONS), meteorological conditions (T and RH), anthropogenic pollutants (O₃, NOX, SO₂ and EC) and OC concentrations observed for Rambouillet (up panel) and Paris (bottom panel). The Pearson correlation coefficients for negative correlation (blue) and positive correlations (red) are presented here. * shows statistically significant (p-value< 0.05) correlation values.**

11. The authors conclude that the forested area of Rambouillet is affected by anthropogenic inputs, influencing the atmospheric chemical composition. A clear quantification or assessment of the relative contributions of anthropogenic versus biogenic sources to the organic aerosol composition at the two sites would help to better understand the extent of the urban influence on the forested area. (Page 19, lines 464-467)

The presence of anthropogenic inputs into the forested site was suggested due to anthropogenic pollutants present at the site, together with the detection of compounds formed from the presence of both biogenic and anthropogenic (organosulfates) emissions. The chemical composition described here at molecular scale was performed by direct infusion analysis and therefore, the peaks associated to the different formulae have a relative intensity, which is not directly proportionally to the quantity of a compound that could be present. Therefore, providing a quantitative contribution of the anthropogenic and biogenic inputs is not possible and it's out of the objectives of this manuscript.

12. The authors state that the HRMS analysis showed similar patterns of the contributions of anthropogenic and biogenic emissions on both sites for periods of pollution. However, a clear definition or criteria for what constitutes a "period of pollution" is missing. More information is needed on how these periods were identified and how they differ from the background conditions. (Page 19, lines 469-471)

Lines 260-268 in the main manuscript are used to define background and pollution periods, which were assigned in function of the chemistry. Background periods represent low OC concentrations together with low levels of the chemistry indicators ($O_3$, $NO_X$ and $SO_2$), while pollution periods are defined for the increase of those. On those periods, the contribution of different air masses was also observed. To clarify this point backward trajectories were calculated for 24 hours with the HYSPLIT model (Stein et al., 2015) using the Global Data Assimilation System (GDAS) as shown in Figure R1. This figure shows examples of air masses trajectories for the Paris area (Rambouillet site) for each of the periods. As observed, during the background period, air masses arrive from the northwest, while in the pollution period, they come from the northeast, crossing urban dense populated areas of Belgium and Paris city. Additionally, during the fire period, air masses from the south of Paris are observed.

*The following text was added into the text to complement the periods definition:*

*Lines 323-328: Backward trajectories were calculated for 24 hours for the Rambouillet site with the HYSPLIT model (Stein et al., 2015) using the Global Data Assimilation System (GDAS) for 1 degree resolution. Those highlights the contribution of different air masses during the background and pollution periods. As observed in Figure S2, during the background period (July 03) air masses arriving from the northwest shows maritime and continental contributions. During the pollution period (July 12), air masses arrive from northeast, crossing urban dense populated areas of Belgium and Paris city.*

*Line 330-331: In agreement, back-trajectories analysis highlight the arrival of air masses from the south of Paris*

[Figure]

**Figure S2. Example of back-trajectories for background and pollution periods calculated with the HYSPLIT model at 1 degree resolution for 24 hours.**

13. The similarities in the chemical composition between the urban and forested sites during the pollution periods have been discussed and the authors are encouraged to supply the potential implications of these findings for the regional air quality and the impact of urban emissions on the surrounding environment. (Page 19, lines 469-471)
The following text was added into the main manuscript:

*Lines 632-634: Similar OA composition observed at the urban and forested areas of the Paris region during such periods reflects anthropogenic and biogenic emissions interactions enhancing summertime aerosol formation. This observation depicts a progressive impact of densely populated areas (megacities) into rural/ forested areas with increasing emissions.*

14. The authors state that the detection of compounds associated with biogenic and anthropogenic oxidation products highlights the importance of understanding urban and rural chemistries at the molecular level. However, they do not provide a clear discussion on how this molecular-level information can be used to improve our understanding of the complex interactions between different emission sources and their impact on the regional atmospheric composition. (Page 19, lines 449-457)
We mentioned that molecular level information improves the understanding of the chemical composition of urban and rural environments as molecular formulae can suggest the possible presence of tracers such as organosulfates $C_5H_{12}SO_7$ (Zherebker et al., 2024). The latest can be associated to specific sources and aerosol formation pathways. For example, in this work, organosulfur compounds were detected, being those formed in the presence of biogenic VOC and anthropogenic pollutants, their presence highlights the mixing between anthropogenic and biogenic emissions. Additionally, as discussed in the manuscript, the approaches mentioned (Van Krevelen diagrams, Aromaticity, families' contribution) provided fingerprints on the organics composition in a megacity and a forested area. The text was rewritten to clarify this point:

*Lines 594-605: The description of the OA chemical composition at molecular scale improves the understanding between the mixing of biogenic and anthropogenic emissions as for example, organosulfur compounds such as $C_5H_{12}SO_7$ and $C_{10}H_{17}NSO_7$ were detected. Those have been previously observed in environments influenced by urban emissions (Kourtchev et al., 2014; Kourtchev et al., 2016; Giorio et al., 2019; Wang et al., 2022) showing the impact of mixed anthropogenic-biogenic air masses. Interactions between different biogenic and anthropogenic components (Rattanavaraha et al., 2016; McFiggans et al., 2019; Shrivastava et al., 2019) were previously reported to influence the OA composition and formation efficiency. Additionally, although only direct infusion analysis was performed in this work, the detection of molecular formulae compatible with those of the literature for possible biogenic (e.g. $C_8H_{12}O_4$ and $C_8H_{12}O_6$) (Kristensen et al., 2013, 2014) and anthropogenic compounds (e.g. $C_6H_5NO_4$) (Iinuma et al., 2010) highlight the importance of providing the chemical composition at molecular scale as organic tracers can be detected. Those can be later associated to specific SOA precursors and formation pathways. As observed, molecular scale information provides fingerprints on the organics composition in the Paris megacity and a forested area.*

15. The authors mention the use of the ACROSS campaign data, but they do not provide any details on the specific objectives, experimental design, or other relevant information about this campaign. (Page 4, lines 110-115)

We have added more details on the description of the ACROSS campaign as suggested by the reviewer.

*Lines 113-119: Atmospheric measurements were performed during the ACROSS campaign, which aims to understand the mixing between the biogenic and anthropogenic emissions and their impact into aerosol formation and aging. In the Paris urban area anthropogenic compounds can be emitted and exported to the urban areas, interacting with their local emissions. Therefore, during ACROSS, measurements were performed at ground-based, airborne, and space-based platforms located at different urban, semi-urban and rural locations in the greater Paris area. Further details of the sites and campaign description are provided in Cantrell & Michoud (2022). In this work, we focus on the aerosol chemical composition measurements performed at ground level at two locations that represent the urban Paris and the suburban forested area.*

16. A big picture or schematic is strongly recommended to summarize the finding in this work.

As suggested by the reviewer a scheme summarizing the findings of this work was added into the supplementary material of this work.

*Lines 456 to 461 were added into the text: Although similar OC concentrations were observed at both sites. We observed differences into the chemical composition, especially at Rambouillet site during the background period. Here, the average conditions highlight the lower number of formulas per aromaticity class for condensed compounds, probably from aromatic origin. While similar percentage of number of formulae were averaged for the pollution periods at both sites (Figure S4) with average values of 40%, 34% and 26% for Paris and 35%, 36% and 29% for Rambouillet of unsaturated, aromatic and condensed classes respectively.*

[Figure]

**Figure S4: Diagram of the summary of the comparison of the aerosol chemical composition at Paris and Rambouillet for samples collected during the ACROSS campaign and analysis performed by thermo-optical method and HRMS.**

Technique Issue:
1. In Figure 2, the y-axis labels for the mass spectra are not clearly legible. Please consider increasing the font size or adjusting the layout to improve the readability of the figure.

The size of figure 2 was adjusted as suggested by the reviewer. Additionally, the colors separation for OC and EC times series were modify to improve readability.

[Figure]

2. Figure S3 are not clearly legible and unreadable.
The transparency of the plots was adjusted to improve readability.

[Figure]

Pereira et al. present analysis of organic aerosols collected during a period in summer in Paris and a forested site outside of Paris. The aerosol was collected on filters during "daytime" and "nighttime" during the course of the sampling period and analyzed offline with a Sunset OC/EC analyzer and high resolution mass spectrometry. Comparison between the two sites in EC and OC mass concentration as well as a select analysis of elemental composition for different days was conducted. Some similarities and differences between the sites was observed. Though the paper may be of interest for the ACP community, there are many aspects the authors need to address prior to publication. Along with the concerns brought up by the other reviewer, which cover many of my concerns as well, the authors need to provide further analysis and description for the following aspects of the paper:

The authors thank to the reviewer for the careful revisions and comments that help us to improve the overall discussion of this manuscript.

1) Methods
1a) It is not clear how the aerosol was sampled onto the filters. Was there an impactor to ensure only PM1 was sampled? Was the high volume aerosol sampler located outside or inside? Was there any sample lines? If there were sample lines, what material was used?
Thank you for the questions. We have increase the description of the sampling into the manuscript to explain these points.

*Lines 137-139 were rewritten as follows: Samples were collected using an automatic continuous high-volume aerosol sampler (30 m3 h-1) DHA-80 (DIGITEL Enviro-Sense) equipped with a $PM_1$ sampling, directly exposed to the ambient air.*
*Lines 151-152 were added into the manuscript: At PEGASUS, gas measurements were performed using Teflon tubes with 6.35 diameter with the lines inlet placed at the top of the container at 2 m.a.g.l.*
*Lines 157-158 were added into the manuscript: For Paris, gas sampling was performed at the top of the building (30 m a.g.l) through a 12 m long Teflon tube, with a 17.5 mm inner diameter at 40 L $min^{-1}$, until a glass manifold where all gas phase instruments sampled ambient air.*

1b) It is stated that the aerosol was sampled at ~20 m above ground in Paris and at ground level at the forested site (line 111 - 112). If the sampler was this low, how much is soil emissions and canopy/below canopy emissions and chemistry impacting the aerosol, compared to the urban site? Can it really be assumed that similar organic aerosol is being observed between the two sites if the sampling is being conducted so far into the forest?
We agree with the reviewer that the height at which the sampling was performed will impact the aerosol chemical composition, especially at the primary fraction of the organic aerosol for the Rambouillet site, where the sampling was performed below the canopy. Therefore, we do not expect to have a similar organic aerosol composition at the urban and forested sites. Although soil emissions and measurements canopy /below canopy were not addresses in this work, we aimed to understand the chemical composition at molecular level at both sites. Following this, we compared the organic aerosol chemical composition at both sites as samples were collected simultaneously and analysed following the same analytical procedure. Additionally, we understand the limitations for the lack of measurement canopy/ below canopy not

addressed in this work, however, we cannot discard the importance of the molecular scale information provided here which allows also to identify the influence of the arrival of urban air masses.

1c) Why were 6:00 - 22:00 and 22:00 - 06:00 time selected for filter sampling? How does this correspond to rush hour or other traffic related emissions as well as other anthropogenic activities, such as cooking? If there was transport from Paris to the forested site, how long is the transport, and how would that impact the aerosol sampled during these times (e.g., how much would the nighttime filter from the forested site be more correlated to the daytime filter from Paris)?

Sampling times during the summer measurements were selected to account for daytime and night time processes. Therefore, the day time filter was collected between 6:00 and 22:00 when sunlight was present and for the night time, filters were collected between 22:00 and 6:00 in the absence of sunlight. Diurnal concentrations profiles for BC and OC previously reported in the Paris urban area (Sciare et al., 2010) showed emission BC peaks for traffic contribution at rush hours between 8:00-10:00 and 20:00-22:00, and OC peaks during the afternoon due to higher photochemistry. Therefore, sampling performed during day time will include an important fraction of anthropogenic activities. Although we do not have an estimation of how long is the transport of the urban air mases to the forested site. The 24 hours modelling with the back-trajectories at 1 degree resolution (100 km) does not allow to accurately separate the two sites by time, however there is ~43 km distance between Paris and the forested area. If we do an approximation of the number of km between Rambouillet and the initial point of the backward trajectory (Belgium), we could have an estimated transport time lower than 4 hours (Figure S2, July 12).

To answer the reviewer concern we have calculated the correlation between the OC concentrations for filters collected at day-time and the corresponding one for Rambouillet collected at night-time. We found a correlation value of 0.55 (p-value=0.02), which may suggest that daily emissions in the urban area can be transported and possibly influence the chemical composition into the forested site during night time measurement. However, other parameters should be taken into consideration, as for example, some oxidation products from monoaromatics compounds have atmospheric lifetimes of 24 hours (Nozière et al., 2015), which suggest that they could be also present in the daytime. More details about the sampling time selection and possible implications have been added into the text:

*Lines 139-142: Sampling times were selected to account for different daily process in the presence of sunlight (day-time) and in the absence of light (night-time). Therefore, sampling performed during day will include an important fraction of anthropogenic activities mainly associated to traffic contribution at rush hours between 8:00-10:00 and 20:00-22:00 as previously reported in the Paris urban area (Gros et al., 2007; Sciare et al., 2010).*

*Lines 289-291: Additionally, at correlating OC concentrations during day time at Paris and night time at Rambouillet, the moderate positive correlation value of 0.55 (p-value=0.02) suggest that some organic compounds could be form in the urban area, transported to the forested one and influence the chemical composition of the consecutive filter.*

[Figure]

**Figure S2. Example of back-trajectories for background and pollution periods calculated with the HYSPLIT model at 1 degree resolution for 24 hours.**

1d) It is not clear in the discussion about extraction how the extractions were being conducted (line 149 - 151). Was methanol used to extract the material, or was it in the slurry ice?

*Lines 168-170 were rewritten to clarify this point: Filters of 46 mm diameter were placed in pre-cleaned glass vials and extracted three times in 3 mL methanol (LC-MS grade, Fisher Scientific) by 30 min sonication in slurry ice.*

1e) What solvent was used for the ESI source (line 155 - 159)? The choice of solvent will impact background and sensitivity/selectivity.

The solvent use in the present study for the ESI source is methanol. We agree that choice of solvent will impact background and sensitivity/selectivity. However, we used methanol due to its convenience and proven efficiency for OA. Previous comparison of the use of other organic solvents demonstrated a lack of differences in molecular composition of extracts when using acetonitrile (LC-MS grade) and methanol (LC-MS grade) (Kourtchev et al., 2013). We have added the following lines on the text:

*The following lines has been added in the text:*
*Lines 171 to 173: Methanol was used due to its suitability for the extraction of polar unsaturated compounds (Zherebker et al., 2024) and high extraction efficiency of OA (Giorio et al., 2019) enabling analysis without extra purification step.*
*Line 182-183: Due to the solvent selection and the use of the ESI source, analysis at molecular scale focus on the polar fraction of the methanol soluble OA.*

1f) It is not clear why CH families were not discuss or report (line 169+). Was this due to type of solvent and/or ionization mode? Similar to how much CHN and CHS may be a large fraction (e.g., reviewer #1

comment), how much is being missed by not reporting CH? E.g., CH would potentially be associated with primary organic aerosol, as discussed in the introduction; however, if the sampling method is biased to not observed primary organic aerosol, how are the results in comparing Paris to the forested site impacted by neglecting this source?

We agree with the reviewer that consideration of hydrocarbons (e.g. from VOC, SVOC) would assist in determining a primary emission from the forest. Yet, in this research we focused on the polar components of OA extractable by methanol. While some of hydrocarbons can be also extracted by methanol, they cannot be ionized by electrospray in negative ionization mode, and would require using of chemical or photoionization, which was out of the scope. Consideration of only a fraction of OA still allowed us to assess the impact of anthropogenic emission on air from the forested area. In the revised version we acknowledged that we analysed only a fraction of the total OA mass.

*Lines 182-183 were added in the manuscript: Due to the solvent selection and the use of the ESI source, analysis at molecular scale focus on the polar fraction of the methanol soluble OA.*

1g) It is not clear nor supported why using O/C and H/C ratios to describe aliphatic and aromatic domains. How justified is this using standards or comparing compounds that may have similar O/C, H/C, and/or elemental ratios but are different between aliphatic and aromaticity? Why isn't the double bound equivalency and/or aromaticity equivalency used instead?

Thank you for the question. We are aware of the limitations of using the O/C and H/C metric as indicators of the oxidation state only as an assignation of aromatic compounds. Therefore, in this study we considered both qualitative analysis, VK visualization and aromaticity equivalent (Xc), which includes DBE in the calculation. We used O/C and H/C ratios only as an indicative of major classes of compounds that have been reported in the literature for the VK patterns (Koch & Dittmar, 2006; Kourtchev et al., 2013, 2016; Mazzoleni et al., 2012; Wozniak et al., 2008). This tool was useful to visualize differences between Paris and Rambouillet samples due to the distribution of possible formulae detected. Then, to better understand the possible presence of aromatic compounds, we used the Xc analysis following Tong et al., (2016) and Yassine et al. (2014).

We have combined Section 3.2.1 and 3.2.2 in the main manuscript to combine both observations and included the following text into the main manuscript to clarify the reviewer point.
*Lines 443-445: Although H/C and O/C values suggested the presence of low oxygenated aromatic compounds (Koch and Dittmar, 2006; Mazzoleni et al., 2012), VK diagrams do not provide further structural information and relying only on H/C and O/C values could be a non-accurate metric for the analysis of aromatic compounds.*

1h) What is condensed aromatics? How is this different from aromatics?

In this work, aromaticity equivalent (Xc) was computed as a qualitative metric to describe the density of double bounds by considering the double bond equivalent (DBE). As suggested in the literature (Tong et al., 2016; Yassine et al., 2014), this approach allows the calculation of aromatic cores and therefore, molecular formulae can account for either aromatic and non-aromatic compounds. Following Yassine et al.

(2014), compounds could be classified aromatics and condensed aromatics in function of the presence of aromatic ring (benzene structure). Aromatics may contain one benzene ring, while condensed aromatics can contain fused rings. Therefore, the threshold values for defining each group were calculated in function of the benzene ring and naphthalene respectively, being the latest considering as the possible simplest condensed structure.

DBE=4, Xc=2.5          DBE=7, Xc=2.7143

As Xc is influenced also influenced by the number of O and S heteroatoms present in the molecule, condensed aromatic structures do not necessarily present more than one aromatic ring.

The following lines (223-224) were added into the text:
*Aromatics may contain one benzene ring, while condensed aromatics can contain fused rings.*

2) Figure 2 is difficult to interpret. For the OC and EC plot, it appears the legend may not match the traces. Also, it is not clear which site is which for wind direction/wind speed. Something that may be useful for Fig. 2 is highlighting time periods where Paris is potentially directly impacted the forested site.

The size and colors of the figure were adjusted as suggested. See reviewer 1 comments for technique Issue, in question 1. The periods of the Parisian emissions impacting the forested site based on wind directions are highlighted in Figure 2. As observed, wind directions arriving from northeast (between 0° and 90°) are more frequent during the second week of July, from 7 to 13 and 15 to 16.

*The following text was added into the main manuscript:*
*Lines 241-242: Wind directions from northeast (between 0° and 90°) shows the impact of Paris emissions into the forest during the second week of July from 7 to 13 and 15 to 16.*

[Figure]

3) Line 222: It needs to be stated that you are comparing the entire sampling period in Paris vs the short period in the forested site. Also, what is the average for Paris during the time period the forested site was sampling for better comparison?

We agree with the reviewer that data for the overlap periods should be consider for better comparison. Therefore, in Table 1 we have included these data and the text has been corrected accordingly. In spite of these clarifications the mean values for Paris considering the whole data set and those corresponding only to the common data with Rambouillet are similar, 3.2 and 3.4 µg m⁻³, respectively.

**Table 1. Summary of OC and EC concentrations observed at Paris and Rambouillet during the summer 2022. Maximum, minimum and mean concentrations are reported for the total data collected from June 14 to July 25 and for the period or data overlap for the two sites (June 27 to July 22). Day and night values of the concentrations are reported for the period common period. The mean concentrations are reported with their standard deviation along the time series.**

| | Paris | | Rambouillet | |
|---|---|---|---|---|
| | OC ($\mu g\ m^{-3}$) | EC ($\mu g\ m^{-3}$) | OC ($\mu g\ m^{-3}$) | EC ($\mu g\ m^{-3}$) |
| All data – mean | $3.2 \pm 1.7$ | $0.4 \pm 0.3$ | $2.9 \pm 1.5$ | $0.2 \pm 0.1$ |
| (min – max) | (0.7– 10.0) | (0.1 - 1.3) | (0.8 - 7.7) | (0.0 - 0.4) |
| Overlap period – mean | $3.4 \pm 1.8$ | $0.5 \pm 0.3$ | $2.9 \pm 1.5$ | $0.2 \pm 0.1$ |
| (min – max) | (1.0– 10.0) | (0.1 - 1.3) | (0.8 - 7.7) | (0.0 - 0.4) |
| Daytime – mean | $3.1 \pm 1.8$ | $0.5 \pm 0.3$ | $2.8 \pm 1.5$ | $0.2 \pm 0.1$ |
| (min – max) | (1.0 – 10.0) | (0.1 - 1.3) | (0.8 - 7.7) | (0.1 – 0.4) |
| Nighttime – mean | $3.7 \pm 1.7$ | $0.5 \pm 0.4$ | $3.1 \pm 1.6$ | $0.2 \pm 0.1$ |
| (min – max) | (0.9 - 7.8) | (0.2 - 1.2) | (1.2 – 5.5) | (0.0 – 0.3) |

*Line 272 was corrected as suggested by the reviewer: mean OC concentrations of $3.2 \pm 1.7\ \mu g\ m^{-3}$ and $2.9 \pm 1.5\ \mu g\ m^{-3}$ were observed for the whole sampling period.*

4) Line 228 - 234: The comparisons here are hard to follow. There are lines where two different concentrations are reported for OC, and it is not clear if it should be OC and EC.

The comparisons in those lines corresponds to OC and EC concentrations and not only OC. We have corrected the error on the text to clarify this point.

*Lines 279: and EC…*
*Lines 281: and EC …*

5) Table 1 and Sect. 3.1.1 -- It is surprising that there is minimal day/ night differences between EC and OC for the sites. How are emissions, chemistry, and boundary layer dynamics playing into impacting these mass concentrations? Why does it appear there is no impact from emissions on EC between day and night?

*The following lines were added into the manuscript:*
*Lines 308 to 309: Higher temperature observed during the day can also enhance SOA precursors emissions such as monoterpenes (Bourtsoukidis et al., 2024; Malik et al., 2023), which then influence photochemical OA formation (P. Lin et al., 2009).*
*Lines 311 to 316: In addition to the influence of temperature, the planetary boundary layer (PBL) dynamics can influence OC and EC concentrations. During the day, the boundary layer height is deeper due to solar heating, which influence the mixing and dilution of pollutants. While during the night, the PBL becomes shallow causing a nocturnal stability, which can enhance pollutants concentrations (Li et al., 2021; Wang et al., 2023; Y. Zhang et al., 2020). Therefore, the lack of tendencies for OC and EC concentrations between day and night can results from a combination of different atmospheric processes.*

6) Sect. 3.2 and Fig. 3 -- Trying to conduct comparisons between the different samples and follow the listing of different suggested molecular identification is difficult here. One potential way to improve comparisons is to either plot the day / night comparison as day with positive values and night as negative values or a scatter plot of day vs night for one site and day vs day, night vs night, and/or day vs night between urban and forested site. This may provide a better way to evaluate similarities and differences between the sites and differences in chemistry.

In section 3.2, examples of mass spectra profiles are used to display the distribution of the different chemical families, together with the presence of some characteristic molecular formulae based on identification previously performed in the literature. However, reducing this analysis to mass spectra or scatter plot comparison will not be accurate as it would depend mainly on the relative intensity (For example, Figure R3). Those intensity values do not represent the actual concentrations of molecular formulae, which additionally can represent the contribution of different isomeric forms.

[Figure]

*Figure R3. Mass spectra comparison for Rambouillet samples collected during day and night on July 03.*

7) Table 2 / Sect. 3.2 -- This table is very uninterpretable. It is difficult to try to compare not only for one site the differences in compound class, but also between urban and forested site. This table may be better for SI and instead some visual way to compare these to better show similarities/differences. Further, though there may be similar CHO (or other classes) across urban and forested, does that really mean that the aerosol is similar? E.g., CHO is dominated by ketones, aldehydes, and/or alcohols in urban area but acids or overall more oxidized material in forested region, but the percent remains approximately the same, it would appear that the organic aerosol is similar when it really is not. Finally, it is surprising how dominant the aromatic/condensed aromatic is in the forested site. Is this all due to transport or something else, as biogenic VOCs/SOA are generally not considered to be aromatic.

Table 2 in Section 3.2 allow us to summarize the compound classes and the aromaticity approach for the different samples selected for HRMS analysis. As suggested by the reviewer, we have included a visual representation of the table for the contribution of the chemical families in Section 3.2 (Figure 4) and remove the table 2. Additionally, the aromaticity analysis was also represented in a visual way at Section 3.2.1

(Figure 5) to clarify this comparison. The values in the text for the family contributions were also updated at excluding CHNS contributions. Table 2 was combined with Table S1 in the supplementary material. Regarding the HRMS analysis, we agree with the reviewer that similar contributions of a family does not necessary represent a similar chemical composition. However, we have provided evidence that highlight possible similar sources for both sites not only due to the families' contribution, but also to similar VK profiles for samples of the pollution period, together with cosine analysis and common molecular formulae identified at both sites.

*Lines 407-409 were rewritten as follows: The fact that similar percentage of molecular classes together with similar OC concentrations are observed in this work, for the periods of pollution at two locations of the Paris region, may suggest similar aerosol sources influencing the chemical composition.*

*Lines 455 to 456: At comparing the variations on the percentage number of formulae for the same site, we observed an increase in the condensed fraction from the background to the pollution site for Rambouillet.*

[Figure]

**Figure 4: Comparison of the percentage of number of formulas per compound class for Rambouillet (RAMB) and Paris samples. Different colors show the compound classes CHO (clear blue), CHON (brown), CHOS (dark blue), and CHONS (orange).**

[Figure]

**Figure 5: Comparison of the percentage of number of formulas per aromaticity equivalent at Rambouillet (RAMB) and Paris samples. Different colors show the compound classes unsaturated (dark blue), aromatic (clear blue) and condensed (yellow).**

8. Fig. 4 is really nice to better represent and emphasize similarities and differences between the sites. Highly recommend diving a little bit more in the discussion/analysis here.

*The following text was added:*
*Lines 427-431: similar patterns at region A for the pollution periods. Similarly, region B (aliphatic) showed a lower density of compounds at Rambouillet during the background periods. A predominance of CHO and CHOS contributions, which can be associated to biogenic or anthropogenic first generation products (Kourtchev et al., 2013), was observed. At Paris, an increase on the density of CHONS compounds in region B compare to Rambouillet was observed, especially for samples collected during the day.*

9. Sect. 3.3 -- This section is very confusing and potential misleading (e.g., correlation does not mean causation). As example, line 389 discusses how relative humidity is negatively correlated with NOx and EC, and this negative correlation is due to RH impacting production of acidic gases, which is not accurate. NOx may produce nitric acid, but it depends on the VOC reactivity, OH concentration, etc. (highly non-linear) and may/may not be correlated with relative humidity. As discussed by Reviewer #1, better description of the chemistry that would lead to the formulas/families provided here and other sections and why there may be correlation is needed.

Thank you for the question. As suggested by the reviewers, we have expanded our explanations for the correlations and the possible mechanisms involved in those (See reply on question 10 for reviewer 1). Regarding the negative correlation between the RH and the EC concentrations, we have not stated that this is a consequence of the acidic gases production "RH was negatively correlated with $NO_X$ (r= -0.82, p-value< 0.001) and EC (r= -0.75, p-value= 0.003) in Paris. This was not the case for Rambouillet, where $NO_X$ and EC levels were lower and instead only negative correlations with $SO_2$ (r= -0.71, p-value= 0.006) were observed. In the presence of humidity $NO_X$ and $SO_2$ gases can transition into acidic species, decreasing their concentration and therefore, being negative correlated with RH". We associate the negative correlation between the species NOx and $SO_2$ with the RH to a transition to acidic species as processes reported in the literature. For example, negative correlations between $SO_2$, $NO_2$ and RH have been previously observed at China (Johnson, 2022; Lin et al., 2019). Lin et al. (2019) associated the negative correlations to conversion process to sulfate and nitrate species, respectively. Johnson (2022) suggested that the increase in RH favors air moisture and rain, increasing the scavenging of those species. We agree with the reviewer that NOx and $SO_2$ are involved in other oxidation processes, such as $NO_2$ with OH, however as additional species were not measure in this work, we do not discuss their influence.

**References**

Atkinson, R. (2007). Rate constants for the atmospheric reactions of alkoxy radicals: An updated estimation method. *Atmospheric Environment*, *41*(38), 8468–8485. https://doi.org/10.1016/j.atmosenv.2007.07.002

Atkinson, R., & Arey, J. (2003). Atmospheric Degradation of Volatile Organic Compounds. *Chemical Reviews*, *103*(12), 4605–4638. https://doi.org/10.1021/cr0206420

Beekmann, M., Prévôt, A. S. H., Drewnick, F., Sciare, J., Pandis, S. N., Denier van der Gon, H. A. C., Crippa, M., Freutel, F., Poulain, L., Ghersi, V., Rodriguez, E., Beirle, S., Zotter, P., von der Weiden-Reinmüller, S.-L., Bressi, M., Fountoukis, C., Petetin, H., Szidat, S., Schneider, J., … Baltensperger, U. (2015). In situ, satellite measurement and model evidence on the dominant regional contribution to fine particulate matter levels in the Paris megacity. *Atmospheric Chemistry and Physics*, *15*(16), 9577–9591. https://doi.org/10.5194/acp-15-9577-2015

Bourtsoukidis, E., Pozzer, A., Williams, J., Makowski, D., Peñuelas, J., Matthaios, V. N., Lazoglou, G., Yañez-Serrano, A. M., Lelieveld, J., Ciais, P., Vrekoussis, M., Daskalakis, N., & Sciare, J. (2024). High temperature sensitivity of monoterpene emissions from global vegetation. *Communications Earth & Environment*, *5*(1), 23. https://doi.org/10.1038/s43247-023-01175-9

Bressi, M., Sciare, J., Ghersi, V., Bonnaire, N., Nicolas, J. B., Petit, J.-E., Moukhtar, S., Rosso, A., Mihalopoulos, N., & Féron, A. (2013). A one-year comprehensive chemical characterisation of fine aerosol (PM2.5) at urban, suburban and rural background sites in the region of Paris (France). *Atmospheric Chemistry and Physics*, *13*(15), 7825–7844. https://doi.org/10.5194/acp-13-7825-2013

Brüggemann, M., Xu, R., Tilgner, A., Kwong, K. C., Mutzel, A., Poon, H. Y., Otto, T., Schaefer, T., Poulain, L., Chan, M. N., & Herrmann, H. (2020). Organosulfates in Ambient Aerosol: State of Knowledge and Future Research Directions on Formation, Abundance, Fate, and Importance. *Environmental Science & Technology*, *54*(7), 3767–3782. https://doi.org/10.1021/acs.est.9b06751

Cantrell, C., & Michoud, V. (2022). An Experiment to Study Atmospheric Oxidation Chemistry and Physics of Mixed Anthropogenic–Biogenic Air Masses in the Greater Paris Area. *American Meteorological Society*, 599–603. https://doi.org/10.1175/BAMS-D-21-0115.1

Chen, Y., & Bond, T. C. (2010). Light absorption by organic carbon from wood combustion. *Atmospheric Chemistry and Physics*, *10*(4), 1773–1787. https://doi.org/10.5194/acp-10-1773-2010

Cheng, Y., He, K., Du, Z., Engling, G., Liu, J., Ma, Y., Zheng, M., & Weber, R. J. (2016). The characteristics of brown carbon aerosol during winter in Beijing. *Atmospheric Environment*, *127*, 355–364. https://doi.org/10.1016/j.atmosenv.2015.12.035

Giorio, C., Bortolini, C., Kourtchev, I., Tapparo, A., Bogialli, S., & Kalberer, M. (2019). Direct target and non-target analysis of urban aerosol sample extracts using atmospheric pressure photoionisation high-resolution mass spectrometry. *Chemosphere*, *224*, 786–795. https://doi.org/10.1016/j.chemosphere.2019.02.151

Gros, V., Sciare, J., & Yu, T. (2007). Air-quality measurements in megacities: Focus on gaseous organic and particulate pollutants and comparison between two contrasted cities, Paris and Beijing. *Comptes Rendus. Géoscience*, *339*(11–12), 764–774. https://doi.org/10.1016/j.crte.2007.08.007

Horník, Š., Sýkora, J., Schwarz, J., & Ždímal, V. (2020). Nuclear Magnetic Resonance Aerosolomics: A Tool for Analysis of Polar Compounds in Atmospheric Aerosols. *ACS Omega*, *5*(36), 22750–22758. https://doi.org/10.1021/acsomega.0c01634

Iinuma, Y., Böge, O., Kahnt, A., & Herrmann, H. (2009). Laboratory chamber studies on the formation of organosulfates from reactive uptake of monoterpene oxides. *Physical Chemistry Chemical Physics*, *11*(36), 7985. https://doi.org/10.1039/b904025k

Iinuma, Y., Müller, C., Berndt, T., Böge, O., Claeys, M., & Herrmann, H. (2007). Evidence for the Existence of Organosulfates from β-Pinene Ozonolysis in Ambient Secondary Organic Aerosol. *Environmental Science & Technology*, *41*(19), 6678–6683. https://doi.org/10.1021/es070938t

Johnson, A. C. (2022). Correlation Study of Meteorological Parameters and Criteria Air Pollutants in Jiangsu Province, China. *Pollution*, *8*(1). https://doi.org/10.22059/poll.2021.321137.1048

King, A. C. F., Giorio, C., Wolff, E., Thomas, E., Karroca, O., Roverso, M., Schwikowski, M., Tapparo, A., Gambaro, A., & Kalberer, M. (2019). A new method for the determination of primary and secondary terrestrial and marine biomarkers in ice cores using liquid chromatography high-resolution mass spectrometry. *Talanta*, *194*, 233–242. https://doi.org/10.1016/j.talanta.2018.10.042

Koch, B. P., & Dittmar, T. (2006). From mass to structure: An aromaticity index for high-resolution mass data of natural organic matter. *Rapid Communications in Mass Spectrometry*, *20*(5), 926–932. https://doi.org/10.1002/rcm.2386

Kourtchev, I., Fuller, S., Aalto, J., Ruuskanen, T. M., McLeod, M. W., Maenhaut, W., Jones, R., Kulmala, M., & Kalberer, M. (2013). Molecular Composition of Boreal Forest Aerosol from Hyytiälä, Finland, Using Ultrahigh Resolution Mass Spectrometry. *Environmental Science & Technology*, *47*(9), 4069–4079. https://doi.org/10.1021/es3051636

Kourtchev, I., Godoi, R. H. M., Connors, S., Levine, J. G., Archibald, A. T., Godoi, A. F. L., Paralovo, S. L., Barbosa, C. G. G., Souza, R. A. F., Manzi, A. O., Seco, R., Sjostedt, S., Park, J.-H., Guenther, A., Kim, S., Smith, J., Martin, S. T., & Kalberer, M. (2016). Molecular composition of organic aerosols in central

Amazonia: Anultra-high-resolution mass spectrometry study. *Atmospheric Chemistry and Physics*, *16*(18), 11899–11913. https://doi.org/10.5194/acp-16-11899-2016

Kourtchev, I., O'Connor, I. P., Giorio, C., Fuller, S. J., Kristensen, K., Maenhaut, W., Wenger, J. C., Sodeau, J. R., Glasius, M., & Kalberer, M. (2014). Effects of anthropogenic emissions on the molecular composition of urban organic aerosols: An ultrahigh resolution mass spectrometry study. *Atmospheric Environment*, *89*, 525–532. https://doi.org/10.1016/j.atmosenv.2014.02.051

Kroll, J. H., & Seinfeld, J. H. (2008). Chemistry of secondary organic aerosol: Formation and evolution of low-volatility organics in the atmosphere. *Atmospheric Environment*, *42*(16), 3593–3624. https://doi.org/10.1016/j.atmosenv.2008.01.003

Lagmiri, S., & Dahech, S. (2023). Weather Types and Their Influence on PM10 and O3 Urban Concentrations in the Cergy-Pontoise Conurbation. *Journal of Applied Meteorology and Climatology*, *62*(5), 549–561. https://doi.org/10.1175/JAMC-D-22-0161.1

Li, Y., Li, J., Zhao, Y., Lei, M., Zhao, Y., Jian, B., Zhang, M., & Huang, J. (2021). Long-term variation of boundary layer height and possible contribution factors: A global analysis. *Science of The Total Environment*, *796*, 148950. https://doi.org/10.1016/j.scitotenv.2021.148950

Lin, C.-A., Chen, Y.-C., Liu, C.-Y., Chen, W.-T., Seinfeld, J. H., & Chou, C. C.-K. (2019). Satellite-Derived Correlation of SO2, NO2, and Aerosol Optical Depth with Meteorological Conditions over East Asia from 2005 to 2015. *Remote Sensing*, *11*(15), 1738. https://doi.org/10.3390/rs11151738

Lin, P., Hu, M., Deng, Z., Slanina, J., Han, S., Kondo, Y., Takegawa, N., Miyazaki, Y., Zhao, Y., & Sugimoto, N. (2009). Seasonal and diurnal variations of organic carbon in PM2.5 in Beijing and the estimation of secondary organic carbon. *Journal of Geophysical Research: Atmospheres*, *114*(D2), 2008JD010902. https://doi.org/10.1029/2008JD010902

Luo, H., Chen, J., Li, G., & An, T. (2021). *Formation kinetics and mechanism of ozone and secondary organicaerosols from photochemical oxidation of different aromatichydrocarbons: Dependence of NOx and organic substituent*. https://doi.org/10.5194/acp-2021-29

Malik, T. G., Sahu, L. K., Gupta, M., Mir, B. A., Gajbhiye, T., Dubey, R., Clavijo McCormick, A., & Pandey, S. K. (2023). Environmental Factors Affecting Monoterpene Emissions from Terrestrial Vegetation. *Plants*, *12*(17), 3146. https://doi.org/10.3390/plants12173146

Mazzoleni, L. R., Saranjampour, P., Dalbec, M. M., Samburova, V., Hallar, A. G., Zielinska, B., Lowenthal, D. H., & Kohl, S. (2012). Identification of water-soluble organic carbon in non-urban aerosols using ultrahigh-resolution FT-ICR mass spectrometry: Organic anions. *Environmental Chemistry*, *9*(3), 285. https://doi.org/10.1071/EN11167

Michoud, V., Hallemans, E., Chiappini, L., Leoz-Garziandia, E., Colomb, A., Dusanter, S., Fronval, I., Gheusi, F., Jaffrezo, J.-L., Léonardis, T., Locoge, N., Marchand, N., Sauvage, S., Sciare, J., & Doussin, J.-F. (2021). Molecular characterization of gaseous and particulate oxygenated compounds at a remote site in Cape Corsica in the western Mediterranean Basin. *Atmospheric Chemistry and Physics*, *21*(10), 8067–8088. https://doi.org/10.5194/acp-21-8067-2021

Mihara, T., & Mochida, M. (2011). Characterization of Solvent-Extractable Organics in Urban Aerosols Based on Mass Spectrum Analysis and Hygroscopic Growth Measurement. *Environmental Science & Technology*, *45*(21), 9168–9174. https://doi.org/10.1021/es201271w

Nozière, B., Kalberer, M., Claeys, M., Allan, J., D'Anna, B., Decesari, S., Finessi, E., Glasius, M., Grgić, I., Hamilton, J. F., Hoffmann, T., Iinuma, Y., Jaoui, M., Kahnt, A., Kampf, C. J., Kourtchev, I., Maenhaut, W., Marsden, N., Saarikoski, S., … Wisthaler, A. (2015). The Molecular Identification of Organic Compounds in the Atmosphere: State of the Art and Challenges. *Chemical Reviews*, *115*(10), 3919–3983. https://doi.org/10.1021/cr5003485

Passananti, M., Kong, L., Shang, J., Dupart, Y., Perrier, S., Chen, J., Donaldson, D. J., & George, C. (2016). Organosulfate Formation through the Heterogeneous Reaction of Sulfur Dioxide with Unsaturated Fatty Acids and Long-Chain Alkenes. *Angewandte Chemie International Edition*, *55*(35), 10336–10339. https://doi.org/10.1002/anie.201605266

Russell, L. M., Bahadur, R., & Ziemann, P. J. (2011). Identifying organic aerosol sources by comparing functional group composition in chamber and atmospheric particles. *Proceedings of the National Academy of Sciences*, *108*(9), 3516–3521. https://doi.org/10.1073/pnas.1006461108

Sato, K., Ikemori, F., Ramasamy, S., Iijima, A., Kumagai, K., Fushimi, A., Fujitani, Y., Chatani, S., Tanabe, K., Takami, A., Tago, H., Saito, Y., Saito, S., Hoshi, J., & Morino, Y. (2022). Formation of secondary organic aerosol tracers from anthropogenic and biogenic volatile organic compounds under varied NO and oxidant conditions. *Atmospheric Environment: X*, *14*, 100169. https://doi.org/10.1016/j.aeaoa.2022.100169

Sciare, J., d'Argouges, O., Zhang, Q. J., Sarda-Estève, R., Gaimoz, C., Gros, V., Beekmann, M., & Sanchez, O. (2010). Comparison between simulated and observed chemical composition of fine aerosols in Paris (France) during springtime: Contribution of regional versus continental emissions. *Atmospheric Chemistry and Physics*, *10*(24), 11987–12004. https://doi.org/10.5194/acp-10-11987-2010

Shrivastava, M., Andreae, M. O., Artaxo, P., Barbosa, H. M. J., Berg, L. K., Brito, J., Ching, J., Easter, R. C., Fan, J., Fast, J. D., Feng, Z., Fuentes, J. D., Glasius, M., Goldstein, A. H., Alves, E. G., Gomes, H., Gu, D., Guenther, A., Jathar, S. H., … Zhao, C. (2019). Urban pollution greatly enhances formation of natural aerosols over the Amazon rainforest. *Nature Communications*, *10*(1), 1046. https://doi.org/10.1038/s41467-019-08909-4

Stein, A. F., Draxler, R. R., Rolph, G. D., Stunder, B. J. B., Cohen, M. D., & Ngan, F. (2015). NOAA's HYSPLIT Atmospheric Transport and Dispersion Modeling System. *Bulletin of the American Meteorological Society*, *96*(12), 2059–2077. https://doi.org/10.1175/BAMS-D-14-00110.1

Surratt, J. D., Gómez-González, Y., Chan, A. W. H., Vermeylen, R., Shahgholi, M., Kleindienst, T. E., Edney, E. O., Offenberg, J. H., Lewandowski, M., Jaoui, M., Maenhaut, W., Claeys, M., Flagan, R. C., & Seinfeld, J. H. (2008). Organosulfate Formation in Biogenic Secondary Organic Aerosol. *The Journal of Physical Chemistry A*, *112*(36), 8345–8378. https://doi.org/10.1021/jp802310p

Surratt, J. D., Kroll, J. H., Kleindienst, T. E., Edney, E. O., Claeys, M., Sorooshian, A., Ng, N. L., Offenberg, J. H., Lewandowski, M., Jaoui, M., Flagan, R. C., & Seinfeld, J. H. (2007). Evidence for Organosulfates in Secondary Organic Aerosol. *Environmental Science & Technology*, *41*(2), 517–527. https://doi.org/10.1021/es062081q

Tong, H., Kourtchev, I., Pant, P., Keyte, I. J., O'Connor, I. P., Wenger, J. C., Pope, F. D., Harrison, R. M., & Kalberer, M. (2016). Molecular composition of organic aerosols at urban background and road tunnel sites using ultra-high resolution mass spectrometry. *Faraday Discussions*, *189*, 51–68. https://doi.org/10.1039/C5FD00206K

Wang, Z., Cao, R., Li, B., Cai, M., Peng, Z.-R., Zhang, G., Lu, Q., He, H., Zhang, J., Shi, K., Liu, Y., Zhang, H., & Hu, X. (2023). Characterizing nighttime vertical profiles of atmospheric particulate matter and ozone in a megacity of south China using unmanned aerial vehicle measurements. *Environmental Research*, *236*, 116854. https://doi.org/10.1016/j.envres.2023.116854

Wei, P., Cheng, S., Li, J., & Su, F. (2011). Impact of boundary-layer anticyclonic weather system on regional air quality. *Atmospheric Environment*, *45*(14), 2453–2463. https://doi.org/10.1016/j.atmosenv.2011.01.045

Wozniak, A. S., Bauer, J. E., Sleighter, R. L., Dickhut, R. M., & Hatcher, P. G. (2008). Technical Note: Molecular characterization of aerosol-derived water soluble organic carbon using ultrahigh resolution electrospray ionization Fourier transform ion cyclotron resonance mass spectrometry. *Atmospheric Chemistry and Physics*, *8*(17), 5099–5111. https://doi.org/10.5194/acp-8-5099-2008

Xu, Z., Feng, W., Wang, Y., Ye, H., Wang, Y., Liao, H., & Xie, M. (2022). Potential underestimation of ambient brown carbon absorption based on the methanol extraction method and its impacts on source analysis. *Atmospheric Chemistry and Physics*, *22*(20), 13739–13752. https://doi.org/10.5194/acp-22-13739-2022

Yan, F., Kang, S., Sillanpää, M., Hu, Z., Gao, S., Chen, P., Gautam, S., Reinikainen, S.-P., & Li, C. (2020). A new method for extraction of methanol-soluble brown carbon: Implications for investigation of its light absorption ability. *Environmental Pollution*, *262*, 114300. https://doi.org/10.1016/j.envpol.2020.114300

Yassine, M. M., Harir, M., Dabek-Zlotorzynska, E., & Schmitt-Kopplin, P. (2014). Structural characterization of organic aerosol using Fourier transform ion cyclotron resonance mass spectrometry: Aromaticity equivalent approach. *Rapid Communications in Mass Spectrometry*, *28*(22), 2445–2454. https://doi.org/10.1002/rcm.7038

Zhang, Q., Jimenez, J. L., Canagaratna, M. R., Allan, J. D., Coe, H., Ulbrich, I., Alfarra, M. R., Takami, A., Middlebrook, A. M., Sun, Y. L., Dzepina, K., Dunlea, E., Docherty, K., DeCarlo, P. F., Salcedo, D., Onasch, T., Jayne, J. T., Miyoshi, T., Shimono, A., … Worsnop, D. R. (2007). Ubiquity and dominance of oxygenated species in organic aerosols in anthropogenically-influenced Northern Hemisphere midlatitudes. *Geophysical Research Letters*, *34*(13), 2007GL029979. https://doi.org/10.1029/2007GL029979

Zhang, Y., Sun, K., Gao, Z., Pan, Z., Shook, M. A., & Li, D. (2020). Diurnal Climatology of Planetary Boundary Layer Height Over the Contiguous United States Derived From AMDAR and Reanalysis Data. *Journal of Geophysical Research: Atmospheres*, *125*(20), e2020JD032803. https://doi.org/10.1029/2020JD032803

Zherebker, A., Babcock, O., Pereira, D. L., D'Aronco, S., Filippi, D., Soldà, L., Michoud, V., Gratien, A., Cirtog, M., Cantrell, C., Formenti, P., & Giorio, C. (2024). Decreasing the Uncertainty in the Comparison of Molecular Fingerprints of Organic Aerosols with H/D Exchange Mass Spectrometry. *Environmental Science & Technology*, *58*(46), 20468–20479. https://doi.org/10.1021/acs.est.4c06749

---

## Author Response (AR2)

We are thankful for the discussion on the manuscript provided by the reviewer. Please, find our responses to each point below.

**Reviewer 2**

Pereira et al. present observations of organic aerosol mass concentration and chemical composition from aerosol collected on filters during a summer field study in Paris. They used EC/OC Sunset and high resolution mass spectrometry for the mass concentration and composition, respectively. The authors found similarities and differences in the mass concentration and composition between Paris and a rural site, suggesting these similarities are related to transport and urban influences on rural chemistry while differences indicate localized chemistry. The authors have done a thorough job addressing concerns from both reviewers.

1. The remaining concern is with Sect. 3.3 and the correlation table. Some of the descriptions are confusing, e.g., line 494 in that CHO compounds can undergo O2 oxidation (unclear how this would be done) or that positive correlation (line 497 - 498) and negative correlations (line 501 - 503) both mean that they were involved in similar chemistry. Similarly, NOx in of itself is not a direct oxidant of gases. I recommend the authors generally soften the language in what the correlations means for these different comparisons and potentially look more into if the correlations actually corresponds to a mechanistic aspect (is T increasing or decreasing the compound through evaporation or chemistry), a chemical reactions (are the compounds produced in photochemistry along with O3), co-emitted, or needed in the reaction (is presence of NO needed to make certain functional groups while the presence of NO may inhibit other functional groups).

Thank you for the comment. We explore and discuss the correlations between the meteorological conditions and chemical composition as a qualitative description. We clarified this point at the beginning and at of the section and rewrite the text to soften some of the observations to address the reviewer concerns:

Lines 449-450 have been added into the manuscript: This sections explores possible correlations and considers the influence of meteorological conditions observed during the campaign on the OA chemical composition.

Lines 465-474 have been rewritten: Positive correlations between $NO_X$ concentrations with the percentage number of molecular formulae for CHON (r= 0.56, p-value= 0.05) and the aromatic subgroup (r= 0.68, p-value=0.01) were observed in Paris. This was not the case of the other chemical families as $NO_X$ was negative correlated with the percentage number of formulae for CHOS, CHONS and unsaturated types. While positive correlations can highlight the role of $NO_X$ into the formation specific groups, the negative correlations can suggest an inhibition effect. Anthropogenic pollutants has been shown to influence the formation of organonitrate compounds (Lim et al., 2016). Also, nitro-aromatic compounds formation in the presence of $NO_X$ was reported by Sato et al. (2022). Possible mechanistic pathways for those groups can derive from CHO compounds oxidation, forming alkylperoxy radicals ($RO_2\cdot$), and subsequent $NO_2$ addition or NO reaction, leading to the formation of N-families (e.g., organo-nitrate, peroxynitrate) (Atkinson, 2007; Kroll and Seinfeld, 2008). The correlations observed are consistent with the different roles of anthropogenic pollutants reported in the literature (McFiggans et al., 2019; Shrivastava et al., 2019).

Lines 512-514 have been added in the manuscript: The correlations observed suggest a potential link between the meteorological parameters, anthropogenic pollutants and the OA composition, however; further research is needed to fully understand the extent of this impact.

2. Further, I recommend the authors follow the authors' guideline for concluding section (https://www.atmospheric-chemistry-and-physics.net/policies/guidelines_for_authors.html), where the results are compared with previous studies and placed into context of this new study, caveats and limitations are addressed, and implications for the results are discussed.

Following the reviewer suggestion, we have added the following lines in the conclusion of the manuscript.

Lines 573–577: These observations provide the first HRMS molecular screening analysis for the Paris region, improving the understanding of OA composition and the differences and similarities between urban and forested areas. However, further information is needed at comparing the chemical composition in different environments in a quantitative way to properly assess the mixing between air masses and their global impact in modelling and air quality studies.